# Exploring the connection between pet attachment and owner mental health: The roles of owner-pet compatibility, perceived pet welfare, and behavioral issues

**Roxanne D. Hawkins**[1]\*, **Annalyse Ellis**[2], **Charlotte Robinson**[3]

1 Department of Clinical and Health Psychology, School of Health in Social Science, University of Edinburgh, Edinburgh, United Kingdom, 2 Department of Psychology, School of Philosophy, Psychology and Language Sciences, University of Edinburgh, Edinburgh, United Kingdom, 3 School of Psychology, Cardiff University, Cardiff, United Kingdom

\* roxanne.hawkins@ed.ac.uk

## Abstract

Research exploring the connection between pet ownership and mental health has expanded substantially in recent years, yet scientific evidence remains inconclusive. Existing studies have oversimplified this relationship by focusing primarily on pet ownership itself, without accounting for crucial factors such as species of the pet, or important relationship dynamics such as owner-pet attachment orientations. This study sought to investigate whether the relationship between owner-pet attachment and owner mental health could be better understood through the lens of owner-perceived pet compatibility, perceived pet welfare, and pet behavioral issues. These under researched aspects are believed to play crucial roles in shaping owner-pet relationships and owner mental well-being. This study surveyed emerging adults (ages 18–26) who owned dogs and/or cats from the United Kingdom (N = 600) who self-identified as experiencing difficulties with anxiety and/or low mood, of whom some reported clinical diagnoses. Our findings revealed that dog owners exhibited more secure pet attachments than cat owners. Anxious attachment was associated with poorer mental health among dog owners, while avoidant attachment was associated with better mental health in both dog and cat owners. Insecure attachment related to poorer perceived pet quality of life, increased reports of pet behavioral problems, and poorer owner-pet compatibility, regardless of pet species. Additionally, poorer welfare and more behavioral problems were associated with poorer mental health for dog owners; these findings were not replicated for cat owners. Notably, a dog's mental state (such as appearing depressed), as well as fear and anxiety, mediated the relationship between owner-pet attachment and owner mental health. Owner-dog compatibility, particularly in the affection domain, positively mediated the relationship between anxious attachment and poorer mental health, while negatively

**Data availability statement:** The datasets presented in this article are not readily available because of participant privacy and ethical considerations. Requests to access the datasets should be directed to the institutional contact ethics.hiss@ed.ac.uk. The project is registered on the OSF platform (link: https://osf.io/s5ejy/).

**Funding:** This project was funded by the Society for Companion Animal Studies (SCAS) Pump Priming Funding Award. Awarded to Dr Roxanne Hawkins.

**Competing interests:** The authors have declared that no competing interests exist.

mediating the relationship between avoidant attachment and better mental health. These findings suggest that a simplistic view of pet ownership fails to capture the complexity of the factors that shape the mental health of pet owners and underscores the need to consider important owner-pet factors to fully understand how the human-pet relationship can impact the well-being of both people and their pets.

## Introduction

Emerging adulthood (ages 18–26), is a distinct and transitional stage bridging adolescence and adulthood [1–3], marked by important decisions regarding consequential areas of life such as education, career, identity and relationships [2] alongside rapid ongoing brain development [4]. This period is also characterized by heightened vulnerability to mental health challenges, including lower life satisfaction, stress, loneliness, and increased risk of psychological disorders [5–9]. Globally, it is the peak age for the onset of mental health difficulties, with approximately 75% of mental health disorders, particularly anxiety and depression, diagnosed between the ages of 18 and 25 years, [6,9,10]. Protective factors such as social support and secure attachments play a critical role in well-being during this time [11–13], with companion animals, hereby referred to as 'pets', representing one potential source of such support.

About 51% of UK households include a pet, most commonly dogs and cats [14]. Pets, particularly dogs, are often acquired for companionship [15], are widely viewed as sources of social support [16], and often play an important role in the daily management of mental health [17]. In emerging adulthood, high importance is often placed onto relationships and meaningful interactions with pets; pets can be self-management tools for emotional health, helping to reduce loneliness and anxiety, and promoting coping and resilience, particularly during hardships and adversity [18–20]. Research suggests pet relationships can support well-being, whereby affection is associated with improved psychosocial functioning [20], and, among LGBTQI+ young adults, greater belonging, positive regard, and emotional support [18]. However, pet ownership can be a source of stress, unmet expectations, and caregiver burden for emerging adults [18,21], reflecting mixed outcomes common in human-animal research across the lifespan [22,23]. Such inconsistencies may stem from the complex interplay of individual and relational factors that can confer both risk and benefit for mental health [24]. Pets may exacerbate distress if owners struggle with caregiving demands or pet health and behavior problems, feel unable to meet their pet's needs, or feel dissatisfied with the relationship [24–27]. Thus, pet relationships can be protective or stressful depending on attachment quality [28–31], perceived compatibility [32], and pet-related characteristics [25,33].

Moreover, pet species differences may also influence well-being outcomes, though findings are inconsistent. Some studies indicate more positive outcomes than negative outcomes for dog owners [34–36], while others found no differences between dog and cat owners [30], with limited research specifically into the well-being outcomes of cat owners [37]. Existing research suggests that dog owners typically

have stronger attachment bonds than do cat owners [35,38], with human–dog relationships characterized by bidirectional and reciprocal emotional engagement that benefits wellbeing [39,40]. In contrast, human–cat relationships may be more variable, with mixed evidence on cats' attachment bonds to humans, and potential different attachment dynamics and attachment behavioral expressions [41,42]. The present research aims to clarify these relationships and their implications for emerging adults' well-being.

### Relationship markers: attachment

Attachment theory, originally developed to describe parent-child relationships, distinguishes between security (feeling assured that one's emotional needs will be met), and insecurity, a two-dimensional construct, consisting of both attachment anxiety and attachment avoidance [43,44]. In human-human relationships, attachment anxiety describes the fear or anxiety that others will not be responsive to emotional needs, while attachment avoidance describes the emotional distance caused by worry regarding others' intentions [44]. Similar features of attachment insecurity have been observed within human-pet relationships, though research has only recently extended attachment theory to this context [45,46]. Measurement remains a challenge: many studies assess bond strength rather than a two-dimensional attachment model. A recent systematic review found that out of 40 studies on attachment and depression, only seven aligned with psychological attachment theory [47]. Furthermore, although pets are often considered close family members and are ranked highly in attachment hierarchies, comparatively little is known regarding attachment within human-pet relationships, compared to human-human relationships [16].

### Pet-related factors: pet behavior and welfare

The present study extends existing research on pet welfare and owner mental well-being, by additionally exploring the role of attachment anxiety and avoidance. Poor pet welfare, preoccupation with potential pet loss, and anticipatory grief may heighten stress and anxiety in owners [26]. Additionally, it is possible that individuals higher in attachment anxiety may feel more positive emotion when engaging in caregiving tasks, and subsequently provide more pet care [48,49]. In contrast, individuals higher in attachment avoidance may find less meaning and feel less positive emotion and thus display reduced caregiving [48–50]. Pet behavior problems can further strain the human–pet relationship, leading to negative emotions (e.g., anger, stress, sadness), decreased life satisfaction, and worse mental health outcomes [25,33,51,52], as well as pet relinquishment and abandonment [33,53]. Moreover, an owner's attachment may influence pet behavior through differences in owner behavioral strategies during challenging situations [54]. Owner attachment may also influence pet behavior as attachment insecurity has been associated with increased dog aggression and separation related disorders [55,56]. Given the potential impact of owner-perceived pet welfare and pet behavior on owner well-being, these overlooked variables were important to consider when examining the relationship between human-pet attachment and owner well-being in the current investigation.

### Relationship markers: compatibility

One important relationship marker, human-pet compatibility, has received little attention in the existing literature, but may play an important role in understanding how attachment influences wellbeing. In human-human relationships, compatibility is a notable component of successful relationships, contributing to higher relationship quality and relationship satisfaction [57,58], which robustly predicts well-being [59]. Similarly, high compatibility, or 'matches' between owners and pets could promote positive functioning human-pet dyads, strengthen bonds, promote relationship satisfaction, prevent pet relinquishment [32], and contribute to better owner mental health [60–62]. However, research into owner-pet compatibility has largely focused on personality matches or mismatches between owner and pet (e.g., [60,63]), overlooking other dimensions (such as physical, emotional, social, and behavioral), that may also shape attachment and well-being and have implications for strategies to promote more successful and functional human-pet dyads. Existing work has further

emphasized dog–owner dyads, neglecting cats. Broader investigation of compatibility across species and dimensions is needed to clarify its role in human–pet relationships and owner well-being.

## The present research

Despite the existing literature related to pet attachment and owner well-being, there are still consequential gaps in the human-animal interaction field's understanding of markers of relationship quality that could be impacting upon owner well-being. This study seeks to address those gaps by answering the following research questions: 1) Are there differences between dog and cat owners on measures of pet attachment and mental health? and 2) Do perceived pet welfare, pet behavioral problems, and owner-pet compatibility, explain the relationship between insecure attachment and owner mental health symptom severity? Based on previous theory and the research findings outlined above, we predict that 1) Dog owners will display more secure pet attachments and thus better mental health than cat owners who will be less securely attached and thus display poorer mental wellbeing, and 2) Pet owners who are insecurely attached to their pets will display poorer mental wellbeing, which will be explained through perceptions of poorer pet welfare, higher rates of pet behavioral problems, and lower scores for perceived owner-pet compatibility [25,29–33].

In this study we focus on anxiety and depression symptom severity for the mental health outcomes. The focus on dogs and cats is justified both by their predominance in UK households, and by the concentration of mental-health research on these species. Even within this domain, however, findings are far from consistent: studies report positive, null, and occasionally adverse effects of dogs and cats on owner well-being. This mixed evidence underscores the need for cautious interpretation of pet-attachment benefits and supports the rationale for our focused scope.

## Method

### Design and procedure

Ethical approval for this study was granted by the School of Health in Social Science Research Ethics Committee (REC) (approval number: 22–23CLPS015). The study was conducted in accordance with the local legislation and institutional requirements. Participants provided online written informed consent prior to participating. Participants were recruited through Prolific, an online recruitment service, a highly efficient recruitment method to ensure high-quality data with a quick turnaround (Prolific, 2014). A balanced sample was requested for an equal distribution of gender. Participants were screened through filters, and the inclusion criteria included: 1) aged 18–26 years, 2) nationality and area of residence was the United Kingdom, 3) first and fluent language was English, 4) had a dog and/or cat, 5) identified as having difficulties with anxiety and/or depression/mood. The UK focus was to ensure a more in-depth and coherent analysis of human-pet relationships within a consistent cultural, legal, and social context, avoiding the complexities and variability inherent in cross-cultural comparisons. Exclusion criteria included: 1) incomplete survey submissions, 2) survey completion times that were three standard deviations below the mean, and 3) failing more than one attention check; in line with Prolific guidance and recommendations.

Data collection commenced 25/05/2023 and ended 09/08/2023. An online survey was hosted on the Qualtrics platform and eligible participants were directed to the survey through their Prolific dashboard. Participants first read an online participant information sheet and provided informed online written consent prior to viewing the survey questions. Once surveys were completed, participants were de-briefed and re-directed to their Prolific dashboard. Given the sensitive nature of the topic, participants were provided with mental health resources both prior to the survey and during the de-brief. The survey took a median length of 14 minutes. Participants received payment using the standard tariff (£3 per 30 mins) which follows ethical pay practices and is in line with the minimum reward per hour reward policy; payments were only made to anonymous participants recruited via Prolific.

## Participants

Priori power analysis: a minimum sample size of $N=68$ was required to achieve 80% power in detecting a medium effect size based on alpha of .05 for each mediational analysis. A total of 656 participants responded to the survey, from this, $n=50$ was returned (i.e., did not meet inclusion or exclusion criteria), and $n=6$ timed out of the survey. The final sample included $N=600$ with $n=341$ dog owners, and $n=259$ cat owners.

Ages ranged from 18–26 years (M = 22.57, SD = 1.90). Some participants had multiple cats (range 1–8, M = 2, SD = 1) and/or multiple dogs (range 1–6, M = 2, SD = 1) but were asked to answer the study questions based on the pet they currently lived with, felt the most attached to, or had owned the longest. Length of pet ownership ranged from under 6 months to over 10 years, and pet age ranged from under 1 year to over 10 years. Most participants lived with their pet full-time at the time of the study ($n=505$), but due to the participant age group, some participants lived with their pet part-time ($n=83$) or did not live with their pet at the time of the study ($n=12$) (e.g., living away from home at university). We aimed to recruit young adults who were having difficulties with their mental health (low mood/depression and/or anxiety); a mental health diagnosis was not required due to low help seeking numbers previously found within this population [64]. Participant demographic information can be found in Table 1.

## Survey measures

Following socio-demographic and pet owning questions, a range of standardized and validated measures were presented, investigating owner-pet attachment, owner-pet compatibility, anxiety, depression, pet behavioral problems, and pet welfare. Some measures were specific to dog or cat ownership.

**Owner-pet attachment.** A limitation of other known pet attachment measures is that they do not align with psychological attachment theory and therefore may not be capturing an attachment relationship. The Pet Anxiety and Avoidance measure [29] was chosen for this study because it was developed from a well-utilized measure for human attachment which taps into the two-dimensional model of attachment in adults, examining both anxiety and avoidant attachment orientations (ECR-R) [65], (RQ) [66]. The measure includes a 16-item scale, each scored on Likert scale from 1–7 ('Strongly Disagree' to 'Strongly Agree'). The measure has two subscales: the Pet Avoidance Scale, and the Pet Anxiety Scale, with 8-items each. Example anxiety items include, "I worry that my pet will stop loving me", and "I fear my pet will abandon me". Example avoidance items include, "I'm uncomfortable being too close to my pet", and "I prefer not to depend on my pet". Scores are continuous, and mean scores are calculated. The measure has been shown to be reliable in previous studies [29]. Cronbach's alphas for this study: Total (α = .83), Anxiety (α = .78), Avoidance (α = .82).

**Owner-pet compatibility.** The Human-Dog Behavioural and Emotional Compatibility (HDBEC) measure is a new measure developed for this study, adapted from González-Ramírez [67]. This measure consists of 20-items to evaluate human preferences ("I enjoy / would enjoy…"), and 20-items to evaluate dog preferences for the same activities ("Your dog enjoys…"), categorized into five domains of compatibility with 4-items each: Physical (e.g., "To exercise with my dog, e.g., running, hiking, walking, swimming"), Social (e.g., "For me and my dog to meet and interact with new people/strangers"), Affection (e.g., "To stroke, pet, and touch my dog"), Closeness (e.g., "To have my dog with me when I relax, e.g., watch tv, read a book"), and Other (e.g., "To take pictures/videos of my dog"). Participants are asked to "Please choose the response that best fits you and your dog" and each item is rated on a 4-point Likert scale: 0 'Totally Disagree' to 3 'Totally Agree'. Scores are compared for owner preference and dog preference for each item, a score of 2 is given for an exact match, and a score of 0 for no match. A total compatibility score is calculated for each domain (range 0–8) and across domains (range 0–40). Higher scores indicate higher compatibility. Cronbach's alphas: Total (α = .90), Physical (α = .78), Social (α = .81), Affection (α = .79), Closeness (α = .81), Other (α = .74). We have made this measure openly available for use for other researchers [https://osf.io/s5ejy/].

**Table 1. Participant demographics.**

| Gender identity | N |
|---|---|
| Male | 297 |
| Female | 278 |
| Non-binary | 23 |
| Prefer not to say | 2 |
| **Sexual orientation** | **N** |
| Heterosexual | 379 |
| LGBTQI+ | 208 |
| Prefer not to say | 13 |
| **Location** | **N** |
| England | 536 |
| Scotland | 42 |
| Wales | 18 |
| Ireland | 4 |
| **Mental health – current difficulties** | **N** |
| Difficulties with anxiety and low mood/depression | 378 |
| Difficulties with anxiety only | 136 |
| Difficulties with depression/low mood only | 83 |
| Prefer not to say | 3 |
| **Mental health – formal diagnosis** | **N** |
| Yes | 325 |
| No | 261 |
| Prefer not to say | 14 |
| **Mental health – when received formal diagnosis** | **N** |
| Recently (less than 6 months ago) | 28 |
| 6-12 months ago | 28 |
| More than 1 year ago | 268 |
| **In romantic relationship** | **N** |
| Yes | 328 |
| No | 272 |
| **Work and education** | **N** |
| Full or part-time student | 246 |
| Working full or part-time | 337 |
| Unemployed and not a student | 97 |
| Other | 15 |

The Human-Cat Behavioral and Emotional Compatibility (HCBEC) measure is a new measure developed for this study, adapted from the HDBEC. The measure structure is the same as the dog measure, also consisting of 20-items for both human and cat preferences, categorized into the same five domains of compatibility: Physical (e.g., "To play with my cat, e.g., games, toys, ball, hide and seek"), Social (e.g., "For my cat to initiate social interactions with me, e.g., nudges me, paws at me, is vocal)", Affection (e.g., "To stroke, pet, and touch my cat"), Closeness (e.g., "When my cat stays close / follows me"), and Other (e.g., "To take pictures/videos of my cat"). This measure is scored and coded the same way as the dog measure. Cronbach's alphas: Total ($\alpha = .90$), Physical ($\alpha = .66$), Social ($\alpha = .66$), Affection ($\alpha = .80$), Closeness ($\alpha = .84$), Other ($\alpha = .75$). We have made this measure openly available for use for other researchers [https://osf.io/s5ejy/].

**Pet welfare.** A direct quality of life (QOL) assessment was included that asked participants "How would you rate your pet's current quality of life?" rated on a scale of 1 "Very poor" to 10 "Excellent". Total scores are calculated.

For dogs, the Canine Health-Related Quality of Life Survey (CHQLS-15) [68] was included that is comprised of 15-items that assess owner perceived dog quality of life through four domains: Happiness (e.g., "My pet enjoys life"), Physical functioning (e.g., "My pet moves normally"), Hygiene (e.g., "My pet keeps him/herself clean"), and Mental status (e.g., "My pet seems dull or depressed, not alert"). Participants are asked to think about the past four weeks when rating each item. Each item is scored on a 5-point Likert scale from 0 "Never / Strongly disagree" to 4 "Always / Strongly agree". Mean scores are calculated for each domain as well as across domains for a total HRQoL score (range 0–4). Cronbach's alphas: Total ($\alpha = .83$), Happiness ($\alpha = .70$), Physical functioning ($\alpha = .70$), Hygiene ($\alpha = .60$), Mental status ($\alpha = .54$).

For cats, the Feline Health-Related Quality of Life (FHQLS) [69] measure first asks participants "Thinking about the past 4 weeks... the general health of my cat has been..?" which they rate on a 5-point Likert scale from 1 "Poor" to 5 "Excellent". Participants then score a further 21-items (e.g., "My cat has yowled in distress") on a 5-point Likert scale from 0 "Not at all / Strongly disagree" to 4 "A great deal / Strongly agree". Negatively worded items are reverse coded and then total scores are calculated; higher scores indicating higher welfare (range 0–89). This measure is comprised of two sub-scales with 8-items each: Healthy behaviors (e.g., "My cat has been bright and alert"), and Clinical signs (e.g., "my cat has been ill or vomited"). Cronbach's alphas: Total ($\alpha = .87$), Healthy behaviors ($\alpha = .69$), Clinical signs ($\alpha = .79$).

**Pet behavioral problems.** For dog behavior problems, The Mini C-BARQ (Canine Behavioral Assessment and Research Questionnaire) [70] was included. The measure is comprised of 42-items that examine owner perceptions of five key domains: Excitability, Aggression, Fear and anxiety, Separation-related issues, Attachment and attention seeking issues, Training and obedience difficulties, and Miscellaneous problems. Each item for each domain is scored on a severity scale of 1–4 (0 = No signs, to 4 = Severe signs), and frequency (i.e., 0 = Never, to 4 = Always). Positively worded items are reverse coded, and then total scores are calculated for each domain and across domains. Cronbach's alphas: Excitability ($\alpha = .61$), Aggression ($\alpha = .85$), Fear and anxiety ($\alpha = .84$), Separation-related behavior ($\alpha = .73$), Attachment and attention-seeking issues ($\alpha = .74$), Training and obedience issues ($\alpha = .60$), Miscellaneous problems ($\alpha = .75$), Total ($\alpha = .89$).

For cat behavioral problems, a measure of owner perceived cat behavioral problems was adapted from Grigg & Kogan [71] and is comprised of 9-items relating to perceived problematic behavior (e.g., destructive behavior, aggression, anxiety/fear, excessive vocalization, house soiling). Owners are asked to report Yes/No for whether their cat shows the specific behavior, and total frequency scores are calculated ($\alpha = .50$). As part of this measure, owners are also asked to rate the degree to which the behavior (if relevant) 'bothers them', on 4-point scale from "Not bothered at all" to "Bothered a great deal". A total score for how much the owner feels bothered by the problems is also calculated ($\alpha = .76$).

**Anxiety symptom severity.** The Generalized Anxiety Disorder Questionnaire (GAD-7) [72] is comprised of 7-items (e.g., "Not being able to stop or control worrying", "Feeling afraid as if something awful might happen"). Participants are asked how often over the last two weeks they have been bothered by the symptoms. Each item is rated from 0 (Not at all) to 3 (Nearly every day). Total scores are calculated, providing a 0–21 severity score ($\alpha = .85$). Scores of 5, 10, and 15 are taken as the cut-off points for mild, moderate, and severe anxiety, respectively. A score of 10 or greater represents a cut-off point for Generalized Anxiety Disorder.

**Depression symptom severity.** The depression module (PHQ-9) from the full PHQ (The Patient Health Questionnaire) [73] is comprised of 9-items (e.g., "Little interest or pleasure in doing things", "Feeling down, depressed, or hopeless"). Participants are asked how often over the last two weeks they have been bothered by the symptoms. Each item is rated from 0 (Not at all) to 3 (Nearly every day). Total scores are calculated, providing a 0–27 severity score ($\alpha = .85$). Cut-off points include scores of 0–4 for no depressive symptoms, 5–9 for mild depressive symptoms, 10–14 for moderate depressive symptoms, 15–19 for moderately-severe depressive symptoms, and 20–27 for severe depressive symptoms.

## Statistical analysis

All analyses were carried out using SPSS 25 (IBM SPSS Statistics for Windows, IBM Corp., Armonk, N.Y., USA) and Hayes' 2013 PROCESS macro for SPSS (V3.5). Prior to conducting the analyses, we tested assumptions of linear regression and mediation analysis, including normality of residuals, linearity, homoscedasticity, and absence of multicollinearity. These diagnostics indicated that assumptions were met within acceptable thresholds. All inferential tests were two-tailed with significance set at $p < .05$. Mediation hypotheses were tested using one-tailed analyses ($\alpha = .05$), informed by a priori theory predicting directional indirect effects.

## Results

### Are there differences between dog and cat owners (IV) on measures of pet attachment and mental health (DVs)?

First, we examined mental health symptom severity within our population. The majority of our participants met clinical cut offs for depression (74.5% of the sample) and anxiety (68.7% of the sample) (Table 2). Then, we examined whether there were differences between dog and cat owners (IV) on measures of mental health and pet attachment (DVs) through independent t-tests. Although dog owners scored higher on anxiety and depression than cat owners, there were no significant differences found (both p > 0.05). Cat owners were more likely to display insecure attachments than dog owners, but a significant difference was only found for anxious attachment (t(492) = −3.13, p = 0.002), and not avoidant attachment (p > 0.05). Descriptive statistics can be found in Table 2.

### Does insecure pet attachment (IV) relate to owner mental health symptom severity (DV)?

Next, we examined whether pet attachment (IV) predicted owner mental health severity (DV) through linear regressions (S1 Table). For dog owners, anxious attachment significantly predicted higher severity scores for anxiety (F(1,340) = 5.85, $p = 0.016$, $r^2 = 0.02$) and depression (F(1,340) = 14.50, p = <0.001, $r^2 = 0.04$). For cat owners, avoidant attachment

**Table 2. Participant mental health symptom severity and pet attachment scores (DVs) based on dog or cat ownership (IV).**

| Depression | N (%) All | N (%) Dogs | N (%) Cats |
|---|---|---|---|
| None | 28 (4.7) | 8 (2.3) | 20 (7.7) |
| Mild | 125 (20.8) | 80 (23.5) | 45 (17.4) |
| Moderate | 186 (31) | 101 (29.6) | 85 (32.8) |
| Moderate – severe | 152 (25.3) | 87 (25.5) | 65 (25.1) |
| Severe | 109 (18.2) | 65 (19.1) | 44 (17) |
| Mean (SD) | 13.65 (5.9) | 13.76 (5.7) | 13.51 (6.1) |
| **Anxiety** | **N (%) All** | **N (%) Dogs** | **N (%) Cats** |
| None | 32 (5.3) | 18 (5.3) | 14 (5.4) |
| Mild | 156 (26) | 90 (26.4) | 66 (25.5) |
| Moderate | 240 (40) | 127 (37.2) | 113 (43.6) |
| Severe | 172 (28.7) | 106 (31.1) | 66 (25.5) |
| Mean (SD) | 11.92 (4.7) | 12.07 (4.8) | 11.72 (4.5) |
| **Anxious pet attachment** | **Mean (SD), range All** | **Mean (SD), range Dogs** | **Mean (SD), range Cats** |
| | 2.47 (.96), 1-5.88 | 2.36 (.86), 1-5 | 2.61 (1.05), 1-5.88 |
| **Avoidant pet attachment** | **Mean (SD), range All** | **Mean (SD), range Dogs** | **Mean (SD), range Cats** |
| | 2.07 (.87), 1-6.25 | 2.04 (.85), 1-6.25 | 2.11 (.91), 1-5.38 |

'Moderate' = cut off for both major depression and generalized anxiety disorder.

significantly predicted lower severity scores for anxiety (F(1,258) = 6.13, p = 0.014, $r^2$ = 0.02) and depression (F(1,258) = 7.06, p = 0.008, $r^2$ = 0.03).

**Does perceived pet welfare (M) explain the relationship between insecure pet attachment (IV) and owner mental health symptom severity (DV)?**

To test assumptions for mediation analysis, we examined relationships between 1) pet attachment and mental health, 2) pet attachment and perceived pet welfare, and 3) perceived pet welfare and owner mental health. First, we examined the relationship between pet attachment and owner mental health (S2 Table and S3 Table). For dog owners, attachment anxiety significantly positively correlated with both anxiety (r = 0.124, p < 0.05) and depression (r = 0.196, p < 0.01) scores. Attachment avoidance significantly negatively correlated with anxiety (r = −0.119, p < 0.05) scores. For cat owners, attachment avoidance significantly negatively correlated with both anxiety (r = −0.190, p < 0.01) and depression (r = −0.177, p < 0.01).

For dog owners, we tested correlations between attachment, perceived pet quality of life (direct assessment scale), total scores and individual subscales on the CHQLS-15, and owner mental health (S2 Table). Anxious and avoidant attachment were both significantly negatively associated with all dog welfare outcomes (anxious attachment: happiness r = −0.405, p < 0.01; physical functioning r = −0.231, p < 0.01; hygiene r = −0.272, p < 0.01; mental status r = −0.242, p < 0.05; total quality of life r = −0.354, p < 0.01; welfare r = −0.219, p < 0.01; avoidant attachment: happiness r = −0.231, p < 0.01; physical functioning r = −0.182, p < 0.01; hygiene r = −0.258, p < 0.01; mental status r = −0.232, p < 0.05; total quality of life r = −0.295, p < 0.01; welfare r = −0.139, p < 0.01). Scores on the subscale 'physical functioning' were significantly negatively associated with owner depression (r = −0.112, p < 0.05) scores. Scores on the subscale 'mental status' were significantly negatively associated with both anxiety (r = −0.153, p < 0.01) and depression (r = −0.163, p < 0.01). Total scores on the CHQLS-15 were significantly negatively associated with depression (r = −0.106, p < 0.05).

For cat owners, we tested correlations between attachment, perceived pet quality of life (direct assessment scale), total scores and individual subscales on the FHQLS, and owner mental health (S3 Table). There was a significant negative correlation between both avoidant and anxious attachment (r = 0.315, p < 0.01), cat healthy behaviors (anxious attachment r = −0.385, p < 0.01; avoidant attachment r = −0.365, p < 0.01) and cat clinical signs (anxious attachment r = −0.275, p < 0.01; avoidant attachment r = −0.163, p < 0.01), total FHQLS (anxious attachment r = −0.375, p < 0.01; avoidant attachment r = −0.294, p < 0.01), and cat welfare (direct QoL; anxious attachment r = −0.321, p < 0.01; avoidant attachment r = −0.159, p < 0.05). There was a significant positive correlation between cat welfare (direct QoL assessment) and anxiety (r = 0.132, p < 0.05).

Next, variables which met required assumptions (i.e., interrelationships existed between them) were further analysed with mediation analysis. First, we examined the mediational effect of a dog's physical functioning (M) on the relationship between anxious attachment (X) and owner depression (Y), and the relationship between avoidant attachment (X) and owner anxiety (Y) (S4 Table). No significant mediations were found; the indirect effects were not significant.

Next, we examined the mediational effect of a dog's mental status (M) on the relationship between anxious attachment (X) and owner anxiety and depression (Y), and on the relationship between avoidant attachment (X) and owner anxiety (Y) (Table 3, Fig 1). Owner-dog anxious attachment had a significant indirect effect on owner anxiety symptom severity through dog's mental status (abcs = .026, large effect); this was a complete mediation as the direct effect of X on Y was no longer significant when accounting for M. Owner-dog anxious attachment had a significant indirect effect on owner depression symptom severity through dog's mental status (abcs = .028, large effect); this was a partial mediation as the direct effect of X on Y remained significant (p = .001) when accounting for M. Owner-dog avoidant attachment had a significant indirect effect on owner anxiety symptom severity through dog's mental status (abcs = .055, large effect); this was a partial mediation as the direct effect of X on Y remained significant (p = .001) when accounting for M.

**Table 3. Parallel mediation analysis examining a) indirect effects of anxious owner-dog attachment (X) on anxiety symptom severity (Y), via dogs mental status (M); and b) indirect effects of anxious owner-dog attachment (X) on depression symptom severity (Y), via dogs mental status (M); and c) indirect effects of avoidant owner-dog attachment (X) on anxiety symptom severity (Y), via dogs mental status (M).**

| | Indirect effects of anxious owner-dog attachment (X) on anxiety symptom severity (Y), via dogs mental status (M) | | | Indirect effects of anxious owner-dog attachment (X) on depression symptom severity (Y), via dogs mental status (M) | | | Indirect effects of avoidant owner-dog attachment (X) on anxiety symptom severity (Y), via dogs mental status (M). | | |
|---|---|---|---|---|---|---|---|---|---|
| | β | SE | 95% CI | β | SE | 95% CI | β | SE | 95% CI |
| Completely standardised indirect effect beta values of X on Y ($ab_{cs}$) (total) | .026* | .012 | .005, .051 | .028* | .012 | .005, .053 | .055* | .021 | .020, .103 |
| Direct effect of X on M (a1) | −.139 | .038 | −.213, −.064 | −.139 | .038 | −.213, −.064 | −.202* | .038 | −.276, −.128 |
| Direct effect of M on Y (b1) | −1.027* | .426 | −1.865, −.189 | −1.322* | .502 | −2.308, −.335 | −1.527* | .433 | −2.378, −.677 |
| Direct effect of X on Y (c`) | .579 | .302 | −.015, 1.174 | 1.158* | .356 | .458, 1.858 | −.879* | .312 | −1.493, −.264 |
| Indirect effect of X on Y via M | .142* | .065 | .026, .282 | .183* | .081 | .033, .355 | .309* | .122 | .112, .581 |

* Significant pathway ($p < 0.05$). Effect sizes: abcs = 0.01 (small effect), abcs = 0.09 (medium effect), and abcs = 0.25 (large effect). M = dogs mental status.

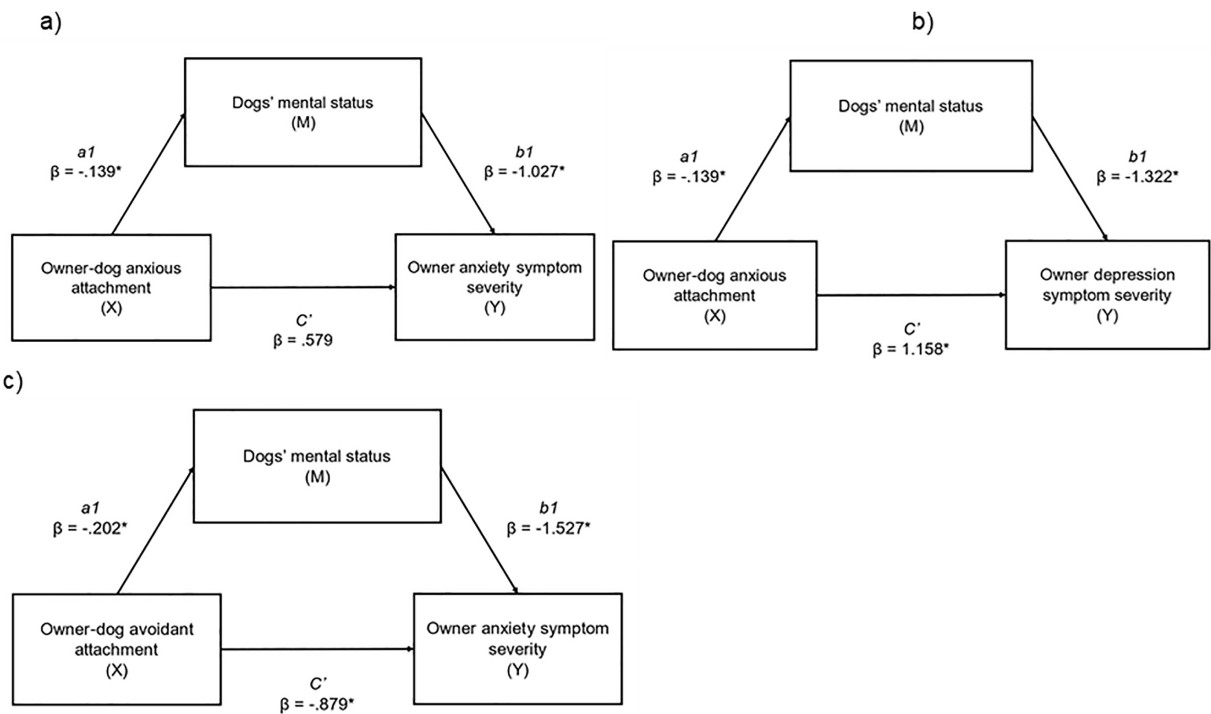

**Fig 1. a) indirect effects of anxious owner-dog attachment (X) on anxiety symptom severity (Y), via dogs mental status (M). (abcs = .026, large effect); b) indirect effects of anxious owner-dog attachment (X) on depression symptom severity (Y), via dogs mental status (M). (abcs = .028, large effect); c) indirect effects of avoidant owner-dog attachment (X) on anxiety symptom severity (Y), via dogs mental status (M). (abcs = .055, large effect). * = significant pathway.**

Next, we examined the mediational effect of perceived dog's welfare (total CHQLS) (M) on the relationship between anxious attachment (X) and owner depression (Y), and the mediational effect of perceived cat welfare (direct QoL assessment) (M) on the relationship between avoidant attachment (X) and owner anxiety (Y) (S5 Table). No significant mediations were found; the indirect effects were not significant.

### Do perceived pet behavioral problems (M) explain the relationship between insecure attachment (IV) and owner mental health symptom severity (DV)?

To test assumptions for mediation analysis, we first examined correlations between owner-pet attachment, perceived pet behavioral problems, and owner mental health. For dog owners, we tested correlations between owner-dog attachment, perceived dog behavioral problems using both totals and subscales of the CBARQ, and owner mental health (S6 Table). There was a significant negative relationship between owner-dog avoidant attachment and dog excitability ($r = -0.147$, $p < 0.01$), and attachment related issues ($r = -0.204$, $p < 0.01$), and a significant positive relationship between owner-dog avoidant attachment and dog aggression ($r = 0.189$, $p < 0.01$) and training difficulty ($r = 0.160$, $p < 0.01$). There was a significant negative relationship between owner-dog anxious attachment and dog excitability ($r = -0.107$, $p < 0.05$), and a significant positive relationship between owner-dog anxious attachment and all other subscales (except for attachment related issues) (aggression $r = 0.182$, $p < 0.01$; fear and anxiety $r = 0.157$; $p < 0.01$; separation issues $r = 0.190$, $p < 0.01$; training difficulty $r = 0.238$, $p < 0.01$; miscellaneous $r = 0.133$, $p < 0.05$) and total issues ($r = 0.148$, $p < 0.05$).

There was a significant positive relationship between scores for dogs' fear and anxiety and owner symptoms of anxiety ($r = 0.146$, $p < 0.01$) and depression ($r = 0.165$, $p < 0.01$). There was a significant positive relationship between dog training difficulty and owner depression ($r = 0.146$, $p < 0.01$).

For cat owners, we examined both the total behavioral problems measure and owners' ratings for 'how bothered' they feel by such problems (S6 Table). There was a significant positive relationship between anxious owner-cat attachment and perceived cat behavioral problems ($r = 0.123$, $p < 0.05$) and a significant positive relationship between anxious owner-cat attachment and ratings of being 'bothered' by such problems ($r = 0.409$, $p < 0.05$). These findings were not replicated for avoidant attachment (S6 Table). Variables which met the required assumptions were further tested with mediation analysis. First, we examined the mediational effect of a dog's fear and anxiety (M) on the relationship between anxious attachment (X) and owner anxiety (Y), and on the relationship between anxious attachment (X) and owner depression (Y) (Table 4, Fig 2). Owner-dog anxious attachment had a significant indirect effect on owner anxiety symptom severity through dog's fear and anxiety ($abcs = .020$, medium effect); this was a partial mediation as the direct effect of X on Y remained significant ($p = .001$) when accounting for M. Owner-dog anxious attachment had a significant indirect effect on owner depression symptom severity through dog's fear and anxiety ($abcs = .021$, medium effect); this was a partial mediation as the direct effect of X on Y remained significant ($p = .001$) when accounting for M.

**Table 4. Mediation analysis examining a) indirect effects of anxious owner-dog attachment (X) on anxiety symptom severity (Y), via dog fear and anxiety (M), and b) indirect effects of anxious owner-dog attachment (X) on depression symptom severity (Y), via dog fear and anxiety (M), and c) indirect effects of anxious owner-dog attachment (X) on depression symptom severity (Y), via dog training difficulty (M).**

| | Anxious owner-dog attachment (X) on anxiety symptom severity (Y), via dog fear and anxiety (M). | | | Anxious owner-dog attachment (X) on depression symptom severity (Y), via dog fear and anxiety (M). | | | Anxious owner-dog attachment (X) on depression symptom severity (Y), via dog training difficulty (M). | | |
|---|---|---|---|---|---|---|---|---|---|
| | β | SE | 95% CI | β | SE | 95% CI | β | SE | 95% CI |
| Completely standardised indirect effect beta values of X on Y ($ab_{cs}$) (total) | .020* | .068 | .012, .273 | .021* | .012 | .002, .050 | .026 | .015 | −.001, .057 |
| Direct effect of X on M (a1) | .129* | .049 | .033, .224 | .129* | .049 | .033, .224 | .196* | .043 | .112, .281 |
| Direct effect of M on Y (b1) | .866* | .331 | .215, 1.517 | 1.068* | .390 | .301, 1.835 | .872* | .442 | .003, 1.742 |
| Direct effect of X on Y (c`) | .610* | .299 | .022, 1.199 | 1.203* | .352 | .510, 1.897 | 1.170* | .361 | .459, 1.880 |
| Indirect effect of X on Y via M | .111* | .068 | .012, .273 | .137* | .084 | .016, .334 | .171 | .097 | −.006, .384 |

*Significant pathway ($p < 0.05$). Effect sizes: abcs = 0.01 (small effect), abcs = 0.09 (medium effect), and abcs = 0.25 (large effect). M = dogs fear and anxiety.

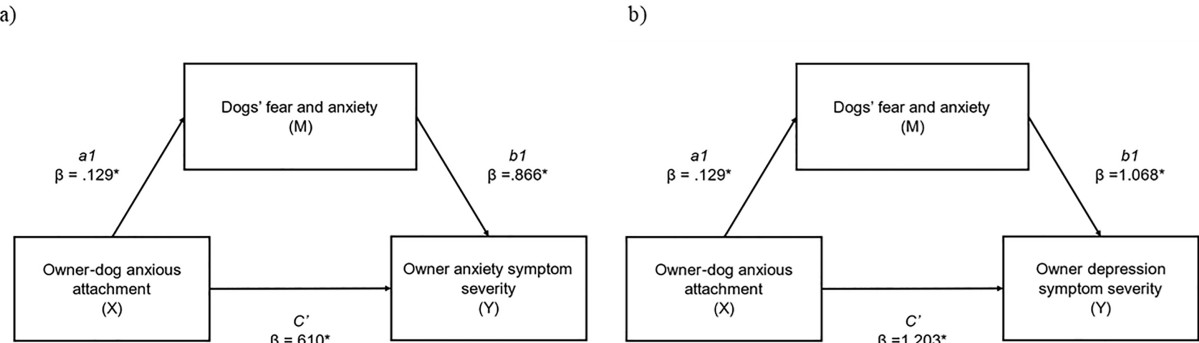

**Fig 2. a) indirect effects of anxious owner-dog attachment (X) on anxiety symptom severity (Y), via dogs fear and anxiety (M). (abcs = .020, medium effect); b) indirect effects of anxious owner-dog attachment (X) on depression symptom severity (Y), via dogs fear and anxiety (M). (abcs = .021, medium effect).** * = significant pathway.

We then examined the mediational effect of dog training difficulty (M) on the relationship between owner-dog anxious attachment (X) and depression (Y) (Table 4). No significant mediation was found; the indirect effect was not significant. A significant positive direct effect was however found for dogs' training difficulty on owner depression; thus, those who reported more difficulties with training their dog, also reported higher depression.

## Does owner-pet compatibility (M) explain the relationship between insecure attachment (IV) and owner mental health symptom severity (DV)?

To test assumptions for mediation analysis, we first examined correlations between owner-pet attachment, perceived owner-pet compatibility, and owner mental health (S7 Table, Supplementary Information). There were significant negative relationships between owner-dog avoidant attachment and all compatibility subscales (except for physical; social r = −0.129, p < 0.05; affection r = −0.325, p < 0.01; closeness r = −0.375, p < 0.01; other r = −0.207, p < 0.01) and total (r = −0.382, p < 0.01) compatibility scores. There were significant negative relationships between owner-dog anxious attachment and all compatibility subscales (physical r = −0.158, p < 0.01; social r = −0.086, p < 0.05; affection r = −0.269, p < 0.01; closeness r = −0.285, p < 0.01; other r = −0.107, p < 0.05) and total (r = −0.308, p < 0.01) compatibility scores. There was a significant positive relationship between the affection subscale and owner anxiety (r = 0.149, p < 0.01).

These analyses were then replicated for cat owners (S7 Table, Supplementary Information). There were significant negative relationships between owner-cat avoidant attachment, subscales for affection (r = −0.277, p < 0.01) and closeness (r = −0.252, p < 0.01), and total compatibility scores (r = −0.246, p < 0.01); thus, those with avoidant attachments felt less compatible with their cats. There were significant negative relationships between owner-cat anxious attachment, and all subscales (except physical and other; social r = −0.225, p < 0.01; affection r = −0.246, p < 0.01; closeness r = −0.273, p < 0.01), and total compatibility scores (r = −0.246, p < 0.01). There were no significant relationships between owner-cat compatibility and owner mental health.

Next, variables which met required assumptions (i.e., interrelationships existed between them) were further analysed with mediation analysis. First, we examined the mediational effect of affection compatibility (M) on the relationship between owner-dog anxious attachment (X) and owner anxiety (Y), and on the relationship between owner-dog avoidant attachment (X) and owner anxiety (Y) (Table 5, Fig 3). Owner-dog anxious attachment had a significant indirect effect on owner anxiety symptom severity through affection compatibility (abcs = −.053, large effect); this was a partial mediation as the direct effect of X on Y remained significant when accounting for M. Owner-dog avoidant attachment had a significant

**Table 5. Mediation analysis examining a) indirect effects of anxious owner-dog attachment (X) on anxiety symptom severity (Y), via affection compatibility (M), and b) indirect effects of avoidant owner-dog attachment (X) on anxiety symptom severity (Y), via affection compatibility (M).**

| | Indirect effects of anxious owner-dog attachment (X) on anxiety symptom severity (Y), via affection compatibility (M). | | | Indirect effects of avoidant owner-dog attachment (X) on anxiety symptom severity (Y), via affection compatibility (M). | | |
|---|---|---|---|---|---|---|
| | β | SE | 95% CI | β | SE | 95% CI |
| Completely standardised indirect effect beta values of X on Y ($ab_{cs}$) (total) | −.053* | .018 | −.912, −.021 | −.042* | .020 | −.083, −.005 |
| Direct effect of X on M (a1) | −.816* | .159 | −1.129, −.504 | −1.003* | .159 | −1.316, −.691 |
| Direct effect of M on Y (b1) | .363* | .100 | .165, .560 | .238* | .104 | .034, .442 |
| Direct effect of X on Y (c`) | 1.018* | .305 | .419, 1.617 | −.331 | .321 | −.961, .299 |
| Indirect effect of X on Y via M | −.296* | .100 | −.502, −.118 | −.239* | .112 | −.465, −.026 |

* Significant pathway (p<0.05). Effect sizes: abcs=0.01 (small effect), abcs=0.09 (medium effect), and abcs=0.25 (large effect). M=affection compatibility.

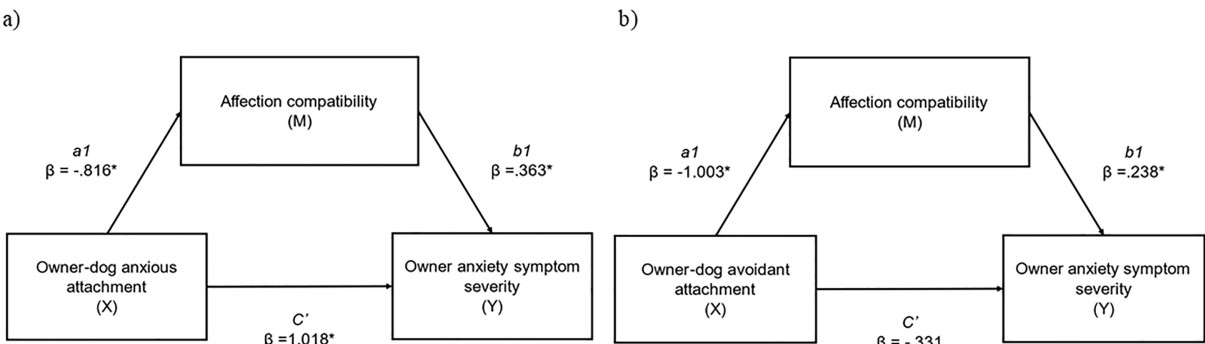

**Fig 3. a) indirect effects of anxious owner-dog attachment (X) on anxiety symptom severity (Y), via affection compatibility (M). (abcs=−.053, large effect); b) indirect effects of avoidant owner-dog attachment (X) on anxiety symptom severity (Y), via affection compatibility (M). (abcs=−.042, large effect).** *=significant pathway.

indirect effect on owner anxiety symptom severity through affection compatibility (abcs=−.042, large effect); this was a complete mediation as the direct effect of X on Y was no longer significant when accounting for M.

## Discussion

This study aimed to examine whether the relationship between owner-pet attachment and owner mental health can be explained by owner perceived pet compatibility, pet welfare, and pet behavioral problems. These aspects, which have previously been under researched, can play crucial roles in shaping owner-pet relationships and can thus consequently influence owner mental well-being [25,29–33]. This study focused on emerging young adults within the United Kingdom who were recruited based on self-identification with current difficulties in anxiety and/or depression. While our study sample was drawn from the general population and a clinical diagnosis was not required, it is noteworthy that a large proportion of the young adults recruited met clinical thresholds for Generalized Anxiety Disorder and/or Major Depressive Disorder, and many exhibited co-morbidity and additional diagnoses. Therefore, our study focused on an underrepresented population within this field of study, and it is important to consider the population when interpreting the findings of this study.

Firstly, our findings revealed no significant differences in mental health symptom severity between dog and cat owners, which could be explained by the high rates of symptoms in both types of owners within our sample. There were differences, however, between dog and cat owners on attachment scores. Dog owners scored higher on pet attachment security compared to cat owners, who exhibited higher attachment anxiety scores. While cat owners also scored higher on avoidant attachment scores, this disparity did not reach statistical significance. These results align with previous research, which has demonstrated that cat owners tend to display more insecure attachments and emotional distance in their pet relationships compared to dog owners who tend to display the reverse [46,74,75]. These findings could possibly be explained by species-typical differences in social behavior. Individuals with avoidant attachment tendencies may seek independence and autonomy in relationships, which could align more closely with the characteristics of a less physically and emotionally demanding pet, one which displays more avoidant attachment-related characteristics, and one in which there are lower expectations from the relationship, such as a cat [74–78]. However, this hypothesis fails to fully account for why cat owners also exhibited higher levels of anxiety in their pet relationships. Anxious attachment typically entails an increased need for reassurance and emotional closeness from relationships, characteristics that might be more readily fulfilled by dogs, given their greater dependence and reliance on their owners [49]. Interestingly, Beck & Madresh [29] also found higher relationship anxiety in cat owners and proposed that cat ownership could be a response to loneliness.

It is important to note, that while the attachment measure used in this study captures attachment orientations (i.e., anxious and avoidant), it may not assess the strength or presence of a true attachment bond. As such, some participants may not have formed a strong attachment to their pet, and species-specific behaviors (e.g., dogs' more overt social bonding) could influence how attachment orientations are expressed and interpreted. Moreover, higher insecurity in human-pet relationships could represent a reduced need for emotional closeness, particularly for those with lower anxiety. Therefore, attachment scores may reflect variation in relationship needs; future research could further explore the functional meaning of differing attachment orientations within human-pet relationships. These findings may also reflect species-specific differences in attachment. While dog–owner relationships are often characterized by mutual emotional dependence, evidence for human–cat attachment is more mixed, which may contribute to observed differences in attachment orientations between cat and dog owners. Further research is needed into attachment-related differences between dog and cat owners before any firm conclusions can be drawn, whilst considering contextual factors such as shared pet responsibilities (e.g., in the context of young adults living at home with their parents). Qualitative investigations could explore individuals' experiences and preferences regarding specific pet species and individual characteristics, considering their attachment orientations. Future research could also expand its scope to include an individual's broader interpersonal relationships, investigating whether attachment orientations observed in human relationships extend to those with pets, or whether pet attachments have unique qualities that can buffer against a lack of secure human attachment, potentially mitigating psychological distress.

Previous hypotheses suggest that individuals facing mental health difficulties may be more inclined to acquire a pet and feel more closely bonded to their pets, actively seeking out pets for emotional support and comfort as a strategy for managing their well-being [17,26]. This notion could potentially shed light on the inconsistencies observed in the field regarding the purported beneficial impacts of pets on mental health [30,79]. However, the reverse has also been proposed whereby secure pet attachment can be a protective factor for mental health difficulties [41,80]. We found partial support for both hypotheses; higher attachment avoidance scores predicted lower anxiety scores regardless of whether a participant had a dog or cat, whereas higher attachment anxiety scores predicted higher depression scores for dog owners only. The findings for anxious pet attachment align with both human-human and human-pet attachment research, indicating that insecure attachment can potentially contribute to poorer mental health with larger associations found for attachment anxiety [79,81,82]. Conversely, the unexpected findings concerning avoidant pet attachment suggest that avoidance might serve as a protective factor against pet owner anxiety. One interpretation is that avoidantly attached individuals may simply be less engaged or emotionally invested in their pets, resulting in fewer bidirectional interaction patterns that

typically drive attachment concerns. Reduced involvement may dampen pet excitability and decrease both the visibility and salience of attachment issues, offering another plausible explanation for the lower anxiety observed in avoidant-attachment owners. Alternatively, it could be posited that avoidantly attached individuals display fears of rejection judgement in their pet relationships in a similar way observed within their close human relationships. In this context, pets can offer a unique form of unconditional and non-judgmental support and acceptance which may serve as a protective buffer against diminished well-being [18,24]. Pets can also provide a secure and non-judgmental avenue for emotional expression and can facilitate emotion regulation for avoidantly attached individuals, who may encounter challenges in expressing and regulating their emotions in interpersonal relationships [82–85]. This could be particularly important for our emerging adult sample who display high mental health symptom severity. This life stage is associated with significant transitions and uncertainty, and societal, psychosocial, and biological factors which can increase psychological distress [5]. These theories need further testing to fully understand the relationship between attachment orientations and mental health outcomes in the context of pet ownership. It is important to also note that the findings reflect associations rather than causal effects and so directionality between attachment insecurity and mental health outcomes cannot be determined from the current study. Furthermore, whether pets are acquired intentionally to support mental wellbeing or whether the perceived mental health benefits emerge after acquisition remains an open question. This could be more effectively explored through future qualitative research, longitudinal data collection, or prospective cohort studies.

We also explored whether perceived pet welfare explained the relationship between insecure pet attachment and owner mental health symptom severity. Our findings indicated that insecurely attached (anxious and avoidant) dog and cat owners were more likely to rate their pet's quality of life lower. An individual's attachment system and caregiving system operate in tandem [43], suggesting that owner-pet attachment could influence how owners perceive and interact with their pets. Previous research supports this theory, demonstrating that stronger pet attachment relates to increased caregiving behavior and positive owner-pet interactions across the lifespan, potentially improving welfare outcomes for pets [48–50]. It has also been proposed in previous research that anxious attachments may in fact foster greater care and attentiveness, whereas avoidant individuals might exhibit more neglectful behaviors [49,50]. It is important to note that our findings may reflect owner *perceptions* of their pet's quality of life rather than the actual welfare of their animals. Insecurely attached individuals often have internal working models that view interpersonal relationships more negatively, leading to heightened worries, concerns, sensitivity to rejection, and challenges in accurately interpreting emotional cues [84–86]. These tendencies might manifest in certain emotional responses to pets, such as attributing negative emotions to them, or misinterpreting ambiguous pet behavior as distress or discomfort, thereby potentially perceiving their pet's welfare as lower, even without substantiated evidence. This may explain why more insecurity in attachment relationships correlated with poorer perceived cat welfare but not clinical signs of health issues. Owner-reported welfare is more subjective and may be influenced by negative appraisal biases, whereas clinical signs may be more objective (e.g., vomiting, limping) and less open to interpretive bias. This hypothesis finds support in qualitative research where individuals with heightened anxiety tend to report maladaptive stress, worries, and anxiety over their pet's welfare. These individuals tend to express feelings of rejection when a pet fails to meet their expectations in a particular interaction, such as not reciprocating physical affection in a time of need [24,26]. Moreover, insecurely attached individuals who perceive themselves as being unable to, or as not currently meeting their pet's welfare needs, may experience feelings of failure, intensifying feelings of insecurity, and exacerbating mental health symptom severity [87,88]. These issues are likely to be particularly pronounced in our sample of young adults experiencing mental health difficulties [24]. Heightened worry about their pet's well-being may exacerbate maladaptive stress responses, caregiving guilt, and emotional overwhelm, further intensifying existing symptom severity and potentially contributing to worsening mental health trajectories.

This study found that for dog owners, higher dog's total quality of life scores correlated with reduced owner depression, while elevated scores on dog's positive physical functioning were linked to decreased owner anxiety. Additionally, higher scores on dog's positive mental status (e.g., happier) were associated with lower levels of both owner anxiety and

depression. These findings are in support of past research highlighting that poor pet quality of life can create psychological distress for owners, relating to poor mental well-being in owners [26,89]. These findings were not replicated in cat owners within our study. We also found no mediational effect of a dog or cat's total quality of life scores on the relationship between attachment and owner mental health. However, a noteworthy finding emerged regarding a complete mediational effect for a dog's mental status. Dog owners who scored higher on attachment insecurity (both anxious and avoidant) reported lower scores for their dog's mental status (e.g., felt their dogs were depressed), which in turn predicted poorer mental health. Cognitive biases commonly observed in individuals with insecure attachment types might influence perceptions of the quality of life of pets whereby negative emotional states can predispose individuals to make more negative judgements about ambiguous social stimuli (in this case a pet's behavior and interactions as indicators of their mental status) thus increasing worry and concern, leading to poorer mental health [90]. This may be especially pertinent in emerging adults whereby cognitive biases associated with insecure attachment may predispose this group to perceive their dog's quality of life more negatively, thereby reinforcing their own distress. Subjective biases in the perception of a pet's quality of life, may have consequences for actual pet welfare [91], highlighting the importance of further investigation.

We were also interested in whether perceived pet behavioral problems explained the relationship between pet attachment and owner mental health. Our findings indicated that cat owners who scored highly on attachment anxiety were more likely to report more cat behavioral problems and were more likely to report being bothered by these problems. Dog owners who scored highly on attachment anxiety were also more likely to report more dog behavioral problems (total scores and all subscales except for attachment issues). This supports our previously proposed hypothesis that attachment insecurity could lead to negative perceptions regarding a pet's behavior and emotional state. Dog owners who scored highly on attachment avoidance were more likely to report lower excitability and fewer attachment related issues for their dogs, supporting our other hypothesis that perhaps avoidantly attached individuals prefer and derive benefits from pets with particular characteristics. However, dog owners who scored highly on attachment avoidance also reported more aggression and training difficulties, suggesting owner-pet attachment could influence dog behavior [55]. This is supported by previous studies also finding an association between high avoidance in dog owners and owner-directed aggression; a theorized explanation being emotional distance, a lack of affection and availability from an owner could result in a lack of perceived secure base for the dog, evoking fear and thus aggression (see [55,56]). Higher reported dog behavioral problems did relate to worse mental health for dog owners in our study, with more dog fear and anxiety relating to increased owner depression and anxiety, and higher dog training difficulty relating to increased owner depression. Moreover, fear and anxiety in dogs partially mediated the relationship between anxious attachment and owner anxiety and depression. These findings support previous hypotheses and evidence that pet challenges could increase burden and influence owner mental health, suggesting that tailored support for pet behavioral issues could alleviate psychological distress [25,33,92]. It is important to note however, that these findings were not replicated for cat owners despite past research demonstrating a link between cat behavioral problems and owner well-being [37]. It is also important to note again that we have focused on perceptions and self-reports of pet behavioral issues which may contain biases, and so these may not reflect accurate depictions of a pet's behavior. Future research could examine how to utilize more accurate assessments of a pet's welfare and behavior to gain a full picture of the possible impact on owner mental health, and what support is needed.

Finally, we explored whether perceived human-pet compatibility explained the relationship between pet attachment and owner mental health. Our findings revealed an association between high scores on attachment insecurity (both anxious and avoidant) and lower perceived total compatibility among both dog and cat owners. Specifically, dog owners who scored highly on attachment anxiety reported lower compatibility across all domains, while avoidantly attached dog owners scored lower on all compatibility domains except for physical compatibility. Similarly, cat owners who scored highly on anxious attachment scored lower on all compatibility domains except for physical and 'other', whereas avoidantly attached cat owners reported lower compatibility specifically in the affection and closeness domains. These findings indicate that insecurity within human-pet attachments could influence feelings of owner compatibility and thereby perceptions of the

human-pet relationship. Alternatively, feeling incompatible with a pet may hinder bonding or foster insecure attachment; given the cross-sectional design, causality cannot be inferred, and the relationship may be bidirectional. Nevertheless, our findings align with our previously proposed hypothesis that insecure pet attachment may foster more negative misattributions of a pet's behavior, which in turn could lead to feelings of emotional disconnection, unmet expectations and needs from the relationship, and relationship dissatisfaction. For example, insecurity within attachment relationships can lead to more negative expectations about a pet's availability and responsiveness, as well as mistrust regarding their intentions [41,42]. In relation to mental health, only the affection compatibility domain seemed to be important for dog owners, yet in reverse to our predictions, with those reporting higher compatibility also displaying higher anxiety scores. Affection compatibility also partially positively mediated the relationship between anxious attachment and dog owner anxiety, and fully negatively mediated the relationship between avoidant attachment and dog owner anxiety. Perhaps those with anxious attachments are more likely to seek out physical proximity and affection from their dogs, and may perceive such efforts as not being reciprocated, heightening feelings of rejection, which increases owner anxiety, whereas those with avoidant attachments do not have the same desire for physical closeness and affection from a pet [45,46]. Placing high value on physical closeness and affection from a pet may indicate a lack of social support from wider human relationships, which can be a risk factor for poorer mental health, yet few studies have simultaneously accounted for the quality of human–human and human–pet relations when considering human well-being [93,94]. Further research is needed to disentangle the complex relationships between pet attachment orientations, owner mental health, and other relationship quality measures including compatibility, with human social support as a potential mediating variable.

A critical drawback of prior investigations into the mental health implications of pet attachment lies in the lack of clear theory-based definitions. Many studies have focused on pet ownership or emotional bond, and do not delineate attachment according to psychological attachment theory or employ measures that reliably evaluate attachment orientations [47]. A notable strength of our study lies in the utilization of standardized theory-driven instruments (ECR-R [65], RQ [66]) that parallel human attachment frameworks [29], thereby enhancing reliability. Our findings show that attachment insecurity relates to poorer owner mental health, and can influence owner perceptions of dog welfare, quality of life, behavior, and perceived relationship compatibility. Future investigations should extend this work by developing age-appropriate measures for younger populations and by examining how attachment orientations interact with bond quality. Although our study did not directly assess bonding satisfaction or depth, factors such as caregiver involvement, interaction frequency, closeness, and shared activities may shape both attachment classifications, bonding, and well-being outcomes. For example, it is plausible that individuals may report strong emotional connections to their pets even when characterized by anxious or avoidant attachment patterns. Accounting for these variables will better contextualize attachment patterns in human–pet relationships.

An additional strength of our study lies in the comparison between dog and cat owners, revealing notable distinctions. However, we did not account for pet age, or duration of pet ownership, both of which may shape attachment and well-being. Attachment to a pet may develop overtime, and the wellbeing implications may be more pronounced at certain time-points. For example, challenges associated with caring for puppies or kittens may delay well-being benefits (e.g., due to heightened stress) until the pet has settled and a bond has formed [24,95], while older pets raise concerns regarding health and anticipated grief [89]. Longitudinal designs could clarify how pet age and length of ownership shape attachment formation and consequent wellbeing benefits.

A limitation of this study is the inclusion of emerging adults who lived at home with family, and some were unemployed, and so our sample may not accurately represent the broader pet-owning population. In such contexts, the young person may not have wanted or chosen the pet themselves, and shared environmental factors (such as living arrangements, finances, autonomy over pet care, and parenting practices) could simultaneously influence both pet attachment and owner well-being. Consequently, our findings may reflect these broader familial dynamics rather than indicating a direct causal relationship between pet attachment and well-being. Future research should explicitly examine these potential

confounding influences, considering the broader social and environmental context of the pet owner. Moreover, our reliance on voluntary participants, and self-identified mental health concerns likely introduced selection bias, further limiting the representativeness of our findings to broader emerging adult populations.

A limitation of our investigation is the absence of data on existing social support and other attachments within participant's human relationships. These factors may moderate the role of pets in mental health; those with strong social support may have their needs saturated and thus have less need to seek support and emotional closeness from their pets [96]. Socio-demographic influences were also beyond our scope but warrant attention. For example, socioeconomic hardship can increase pet burden while limiting access to support, thereby compounding mental health risks [97]. Personal characteristics such as identity are also important. In our sample, 35% identified as being LGBTQI+, a group disproportionately vulnerable to distress due to stigma and discrimination, yet one in which pets may serve as vital sources of resilience and stress reduction, playing an outsize role in buffering against some of these hardships [98]. Moreover, gender identity [16,99] and owner personality characteristics [32,60], may further shape human-pet dynamics. Finally, other relational concepts, such as self-expansion, perceived pet responsiveness, and perceived pet insensitivity [30] remain underexplored and could further illuminate how pet relationships influence owner well-being.

While our focus has been on dogs and cats, broader evidence suggests that pet effects vary by species, implying that not all companion animals confer equivalent mental-health advantages. Future research should examine if similar mechanisms hold for other common pets such as rabbits, birds, and small mammals. Furthermore, the number of pets an individual owns may moderate both emotional support and caregiving burden, particularly when resources for pet care may be scarce. Future studies are required to clarify whether multiple pets enhance mental-health outcomes or increase stress. Additionally, expanding this work to non-university-age populations, cross-cultural settings, and individuals without diagnosed mental health conditions will enhance generalizability and guide future pet-inclusive wellbeing initiatives. Finally, given the number of analyses conducted, the risk of Type I error is elevated. Although results were theoretically informed, future confirmatory studies should apply more stringent corrections (e.g., Bonferroni) or use pre-registered hypotheses to mitigate this risk.

## Conclusion

This study reveals the complex interplay between pet ownership, owner–pet attachment, and mental health, revealing meaningful differences between dog and cat owners. Our findings emphasize the critical role of multidimensional owner–pet compatibility, spanning physical, emotional, social, and behavioral congruence, in shaping attachment quality and psychological outcomes. Notably, individuals with elevated attachment anxiety may derive emotional satisfaction from caregiving, reinforcing both caregiving behaviors and attachment strength. Conversely, when pets exhibit welfare issues or behavioral problems, real or perceived, owners often experience increased stress, guilt, anxiety, and diminished well-being. Interventions must therefore not only promote compatibility across key domains but also deliver structured support for such owners. This could include behavioral training, psychoeducation regarding species-typical norms, and mental health resources to alleviate caregiver burden and rebuild positive relational dynamics. From a clinical perspective, targeted programs addressing expectation management, compatibility-based pet selection, and attachment-related vulnerabilities, such as offering structured support for anxious individuals to harness caregiving satisfaction without fostering dependency, are warranted. At the service-delivery level, mental health providers and university support services might implement home or campus-based pet assisted interventions grounded in attachment principles and compatibility assessment. Such tailored interventions, capable of mitigating cognitive biases in anxious individuals while fostering relational engagement for avoidant individuals, may hold promise for enhancing both owner mental health and pet welfare, ultimately sustaining healthier and more functional human–pet dyads.

## Supporting information

**S1 Table. Linear regressions for insecure pet attachment (IV) predicting owner mental health severity (DV).**
(DOCX)

**S2 Table. Spearman two-tailed correlations between pet attachment and perceived dog welfare, and between perceived dog welfare and owner mental health.**
(DOCX)

**S3 Table. Spearman two-tailed correlations between pet attachment and perceived cat welfare, and between perceived cat welfare and owner mental health.**
(DOCX)

**S4 Table. Parallel mediation analysis examining a) indirect effects of anxious owner-dog attachment (X) on depression symptom severity (Y), via dogs physical functioning (M), and b) indirect effects of avoidant owner-dog attachment (X) on anxiety symptom severity (Y), via dogs physical functioning (M).**
(DOCX)

**S5 Table. Mediation analysis examining a) indirect effects of anxious owner-dog attachment (X) on depression symptom severity (Y), via dogs total quality of life scores (CHQLS) (M), and b) indirect effects of avoidant owner-cat attachment (X) on anxiety symptom severity (Y), via cats quality of life scores (direct assessment) (M).**
(DOCX)

**S6 Table. Relationships between owner-pet attachment, perceived pet behavioral problems, and owner mental health.**
(DOCX)

**S7 Table. Relationships between owner-pet attachment and owner-pet compatibility (subscales and total score), and between owner-pet compatibility and owner mental health.**
(DOCX)

## Acknowledgments

A sincere thank you to all of the young adults who took their time to participate in this research.

## Author contributions

**Conceptualization:** Roxanne D. Hawkins.

**Data curation:** Roxanne D. Hawkins, Charlotte Robinson.

**Formal analysis:** Roxanne D. Hawkins.

**Funding acquisition:** Roxanne D. Hawkins.

**Investigation:** Roxanne D. Hawkins, Charlotte Robinson.

**Methodology:** Roxanne D. Hawkins, Charlotte Robinson.

**Project administration:** Roxanne D. Hawkins, Charlotte Robinson.

**Resources:** Roxanne D. Hawkins.

**Software:** Roxanne D. Hawkins.

**Supervision:** Roxanne D. Hawkins.

**Validation:** Roxanne D. Hawkins.

**Visualization:** Roxanne D. Hawkins.

**Writing – original draft:** Roxanne D. Hawkins, Annalyse Ellis, Charlotte Robinson.

**Writing – review & editing:** Roxanne D. Hawkins, Annalyse Ellis.

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
