## [Decision Letter · Decision Letter 0]

13 Feb 2025

PONE-D-24-52893
Exploring the Connection Between Pet Attachment and Owner Mental Health: The Roles of Owner-Pet Compatibility, Perceived Pet Welfare, and Behavioral Issues.
PLOS ONE

Dear Dr. Hawkins,

Thank you for submitting your manuscript to PLOS ONE. After careful consideration, we feel that it has merit but does not fully meet PLOS ONE’s publication criteria as it currently stands. Therefore, we invite you to submit a revised version of the manuscript that addresses the points raised during the review process.

We look forward to receiving your revised manuscript.

Kind regards,

Janak Dhakal

Academic Editor

PLOS ONE

Journal Requirements:

3. Thank you for stating the following financial disclosure: [This project was funded by the Society for Companion Animal Studies (SCAS)

Pump Priming Funding Award. Awarded to Dr Roxanne Hawkins.]

4. In the online submission form you indicate that your data is not available for proprietary reasons and have provided a contact point for accessing this data. Please note that your current contact point is a co-author on this manuscript. According to our Data Policy, the contact point must not be an author on the manuscript and must be an institutional contact, ideally not an individual. Please revise your data statement to a non-author institutional point of contact, such as a data access or ethics committee, and send this to us via return email. Please also include contact information for the third party organization, and please include the full citation of where the data can be found.

Additional Editor Comments:

ABSTRACT

1. L-52, instead of using ‘overlooked’ in past research, I would use soft tone words.

INTRODUCTION

1. I understand that bring human-human relation and psychology to compare and contract the human-pet relation is important, however the introduction section is too long. I recommend to trim it down. Having said so, I think more could be introduced about dog versus cat ownership and attachment.

2. When discussing about such important subject, please provide recent data on pet ownership, cat and dog ownership stats., etc. The following two manuscripts have cited that information.

https://doi.org/10.1111/1541-4337.70060

https://doi.org/10.1016/j.cvsm.2021.01.010

METHOD

1. L273 and L434-435-Just curious why being a UK nationality was important for this study?

2. How was the participants selected that have such a huge proportion of people who were having mental health difficulties? Is this a random sample or selective/targeted samples?

DISCUSSION

1. Did the study find an answer to a question on whether, the pet owners in the study owned pets to seek mental wellbeing or they realized that the pets became part of their mental health, once they started keeping pets?

Reviewers' comments:

Reviewer's Responses to Questions

**Comments to the Author**

1. Is the manuscript technically sound, and do the data support the conclusions?

Reviewer #1: Yes

Reviewer #2: Yes

2. Has the statistical analysis been performed appropriately and rigorously? 

Reviewer #1: Yes

Reviewer #2: No

3. Have the authors made all data underlying the findings in their manuscript fully available?

Reviewer #1: Yes

Reviewer #2: Yes

4. Is the manuscript presented in an intelligible fashion and written in standard English?

Reviewer #1: Yes

Reviewer #2: Yes

5. Review Comments to the Author

Reviewer #1: This study explored the interaction of pet attachment orientations and owner wellbeing in emerging adults owning cats or dogs, while considering the role of pet compatibility and perceived pet wellbeing and (problem) behaviour. They found an intricate interplay between the factors, with insecure pet attachment correlating differently with anxiety or depression in dog vs cat owners. Moreover, compatibility and behaviour mediated these interactions in both species-owners, while only dogs’ mental wellbeing had a mediating effect among the perceived pet wellbeing factors. The authors conclude that the dynamics of pet ownership and wellbeing are much more complicated than generally assumed, and that pet attachment might be a risk factor for wellbeing.

Given that more than half of the population is estimated to own a pet and that the literature on pet ownership and wellbeing is divided, the paper tackles an important topic and provides a novel investigation by combining factors that have previously shown to influence human and animal wellbeing. Furthermore, they focus on a demographic that is particularly vulnerable to mental health challenges. The paper provides a well-structured introduction into the different factors they plan to test and the scales they used to do so. The results are likewise well structured and in combination with the figures, easy to follow. However, I point out some caveats in the interpretation of the results and the appraisal of the methods that might benefit the validity of the paper.

In short, I see three main issues: 1) the methodology (self-reports) only allow for correlations, and while the mediation analysis has some merit, I do not feel that the conclusions and interpretations can be stated as causally as they were. For example, the results underlying the conclusion that “attachment insecurity [is] a potential risk factor for poorer owner mental health” might equally be explained by people with poorer mental health forming more insecure attachments to their pet, given that we do not know based on the questionnaire whether the animal or the mental health problems came first. 2) the authors measure whether the participants were more or less avoidantly/anxiously attached to the animals, but there was no measure on how strong that attachment was (e.g. some score on bonding strength) or whether it was an attachment (in the deeply bonded sense of the word) at all. This might not only have differed across participants, but also between cats and dogs, given that we know dogs exhibit attachment behaviour, but cats might not (references see introduction). While this cannot be amended in hindsight, I would be grateful to hear the authors’ opinion on how this might or might not have influenced the attachment orientations, and if needed, discuss this drawback as limitation. 3) Finally, I am missing consideration for the study populations’ background factors, especially the fact that most of the people still lived at home. For one, this might suggest that many of them did not choose their pet themselves and/or have a different relationship to the pet than single/pair pet owners do (e.g. fewer caretaker responsibilities, less choice on getting a compatible animal, less involvement in training/daily interactions…). Secondly, this might mean that family environment (e.g. parenting style) might influence both the pet attachment and the owner wellbeing rather than one influencing the other directly (see details below). While they are a critical demographic, they may hence not be the most representative model to generalize the outcomes to pet ownership in general. Most points in the discussion could benefit from these considerations. Beyond that, I point out some minor issues like lacking consistency or unclear sentences below. Beside these interpretational points, the paper has tackled an important topic with sound methodology, and I see merit in its publication after the consideration of the abovementioned caveats.

General:

- Please be consistent in language use, e.g. well-being and wellbeing, behaviour and behavior.

Introduction

- L133 more positive compared to other pets? Or more positive rather than negative? And from what I know, cats are among the most studied pets after dogs. What about bunnies, rats, birds, reptiles? If they are not of interest, I would recommend either specifying the general pet comment or adding why you focus on these two species in particular (e.g. “dogs and cats are the most frequent pets and even there, outcomes are ambiguous or lacking.”)

- L135 “Human-dog attachments are, however, bi-directional and there is reciprocity within the relationship, which is important for wellbeing [35, 36], yet less is known regarding the nature of human-cat relationships, which may be more variable, and where different attachment features may be observed [37, 38].” I have trouble following this argument. Partially, it repeats what it says above. Partially, I am confused about the “attachment features”. Do you mean it might not be bidirectional with cats? Studies diverge on whether cats form attachment to humans or not, while the case is clearer for dogs. Mentioning these might support your argument. But you don’t bring it up again in the discussion, so I am not sure what you really mean here.

o Edwards, C., Heiblum, M., Tejeda, A., & Galindo, F. (2007). Experimental evaluation of attachment behaviors in owned cats. Journal of Veterinary Behavior: Clinical Applications and Research, 2(4), 119–125. https://doi.org/10.1016/j.jveb.2007.06.004

o Potter A, Mills DS (2015) Domestic Cats (Felis silvestris catus) Do Not Show Signs of Secure Attachment to Their Owners. PLOS ONE 10(9): e0135109. https://doi.org/10.1371/journal.pone.0135109

- L147 looks like a double space

- L159 operationalizing?

- L160-166 This is a repetition of what was said above. For a clearer reading experience, I would recommend focussing the last sentence on the family member part, rather than repeating the difference between well-studied human attachment and lack thereof with pets again.

- L175f I am curious. Would you expect secure vs insecure attachment to be correlated to high vs low bonding, given that there is more dissatisfaction in the bonding experience when the attachment is insecure?

- L180 “…we can hypothesise that similar psychological benefits may be observed within secure human-pet attachments.” Unfortunately I do not follow this argument. Which “similar psychological benefits”? Similar to what? The ones in secure human-human attachments? These have not been discussed yet. But there is evidence that more secure attachment with pets is associated with better psychological wellbeing (L175-177). Plus, the secure base effect you mention was likewise only found in less insecurely attached owners (see Zilcha-Mano et al 2012, [42]). In other words, there is already evidence for that, why bring it up as a new hypothesis? Or do you mean that you expect further psychological benefits that mirror most of the ones found in secure human-human attachments, beyond the ones that were already found in owner-pet studies? Then naming these more specifically might be helpful.

- L185 upon

- L192 well-being or wellbeing (L186). Please be consistent

- L220 behaviour or behaviour (L61) (also within abstract)

- L252 underlooked

Methods:

- I am missing a statistical method section. You mention the analysis in the result section but did you check the model assumptions, and if yes, which ones? What programme did you use? What packages? Accrediting the developers of these helpful softwares/packages is a nice way to thank them for their work.

Results:

- L451 You spoke of associations between wellbeing and attachment in the attachment paragraph in the introduction as well as in the research question, but switch to more causal relationship (“predicted”, use of linear regression rather than correlation) here in the results. I do recognize that you do not specifically claim a causal uni-directional link here in how you present the results. But I am curious why you chose a linear regression over a correlation? Without causal studies and given that we are talking about home-living young adults, I would be surprised if the pet attachment majorly played into the wellbeing rather than both being a result of e.g. home live, parental strategies, etc. I could even imagine that the causality is reversed in some cases, with more anxious people developing more anxious relationships with their new pets (or, given the self-report measures, currently perceiving their attachment as more anxious). Using a correlation might hence be more suitable.

- L454-457 the last sentence is redundant to the previous one, particularly since it suggests a more uni-directional relationship (for comment on that see L451)

- Table 7: delete “examining a) indirect effects” in line 2, the “a)” is double

Discussion

- L 907 “it is” rather than “it’s”

- L911 “is IS”

- L921 – 940 These are a range of interesting explanations. However, I suggest that they might be overlooking an inherent factor of the study population: The vast majority of the emerging adults still lived at home. Hence, the choice of the pet might not have been theirs (or at least not theirs alone), and the ownership of a family animal is also different from an animal one gets for oneself. Especially in the case of cats, this might explain the higher insecure attachment style. Anxious, if one would have liked a pet that one can closely bond to and interact with, but the family chose for it to be a cat, and a cat might not be what meets one’s needs. And avoidant (though n.s. here), because the caretaking/interaction time needed for a cat is already a lot lower than with dogs and given that is shared across the family, the bonding and hence emotional connection might be on a lower level than with a dog.

o On that note, I am wondering in how far the relationship the participants have with their animals can truly be understood as attachment. From what I can see, there was no measure in the questionnaire how much time the participants spent taking care of/ interacting with the animals, and whether they were attached (in the psychological sense of the word) at all. E.g. a comparison how strongly people were bonded to cats vs dogs would have been helpful. Attachment is usually reserved for deep connections such as child-caregiver or romantic relationships and takes a long time and many emotional interactions to form. I am therefore not entirely convinced whether the relationship one has with a family pet is, in all cases, classified as an attachment bond. Did you take this into consideration at some point in the questionnaire/ recruitment process? And if not, how do you reckon this might have influenced the distribution of secure/insecure attachment? E.g. it might explain why cats’ wellbeing or behavioural problems did not correlate with the owners’ wellbeing measures.

- L941 are these your hypotheses or are they from someone else? A clearer attribution would be helpful in either case.

- L948 not regardless of pet species, but regardless of dog or cat

- L953 or, perhaps people who are less anxious don’t feel such a need to deeply bond with their animal, which might reflect as a lower emotional connection in the attachment test (if not filtered for no/low attachment beforehand)

- L969-993 Thank you for brining this up and explaining it well. It is a very important consideration in this kind of self-reported research and I am glad you considered it.

- L1022 or perhaps they are less involved/not overly focused on their pet, leading to less excitability in the animal and lower focus on attachment issues (because avoidant-attachment people might not be as sensitive to lacking attachment)?

- L1052 Again, I am wondering here whether the relationship could be the other way around. Meaning once the family got the cat, the participant was not well compatible with it and hence did not develop a secure attachment.

- L1092 Given the correlational nature of your data, I find this conclusion too definitive. Your finding is very interesting and potentially important in that attachment insecurity correlates to low wellbeing, but calling it a risk factor would suggest a definitive causal relationship which has not been tested for with the necessary long-term studies (e.g. pre/post pet adoption). I would suggest being a bit more conservative here.

Reviewer #2: The article titled “Exploring the Connection Between Pet Attachment and Owner Mental Health: The Roles of Owner-Pet Compatibility, Perceived Pet Welfare, and Behavioral Issues” by Hawkins et al. investigates the relationships among attachment style of pet owners, mental health status, mood, pet behaviors, and species differences.

I found the paper interesting. I liked it, but I had great difficulty in extracting the major findings, because of how the information was presented. Therefore, I have the following minor and major suggestions on how to better communicate the conceptualization and the results of this paper.

A. Minor Issues and Suggestions:

1. It is better to call certain type of nonhuman animal behaviors, such as anxiety as “anxiety-like” , because these are human attributes, And they should be defined more clearly (see abstract and other places)

2. Line 97, ref 10, perhaps it would be better to specify location of these findings. This finding wouldn’t (shouldn’t) apply globally.

3. Lines 114 through 122: requires a re-write, especially for line 117. What is substituting-related attachment, for example?

4. Line 135-137, expand bi-directional nature of dog-human relationship

5. Line 151-156: references needed in this section

6. Line 162: how do we define human wellbeing outcomes? Give examples.

7. Line 270. Provide reference (for Prolific) in here.

8. Line 320: Owner-pet attachment: a few examples of questions are needed here just like the owner-pet compatibility section.

9. Line 459 Table 3, add alpha (in general in the beginning of stat tests) and whether this is one or two tailed test). I assume betas are standardized coefficients, because the authors say the strength of the relationship was also measured. If so, state it explicitly.

10. Line 941, are these hypotheses yours? If not, state references here.

11. Line 1057 and anywhere relevant: use the term hypothesis instead of theory.

B. Major Issues and Suggestions:

1. There are 3 hypotheses presented in Intro, but 5 in Results. There should be an exact correspondence between sections, not only in terms of the numbers but also wording.

2. Many paragraphs in the Intro belong to Discussion, perhaps half of the section.

3. Intro, as a result, should significantly be revised. The authors should state the current state of our understanding on this topic with gaps in our knowledge, then how they are planning to close this gap. And, then make predictions and support your predictions with a rationale coming from previous research. The present research stated at the end should be the authors' anchor in writing the intro. Remove all these sections. They don’t belong to Intro. They are confusing. Intro should be crisp.

4. Each hypothesis in the intro should then be specifically addressed and expanded with implications in the discussion.

5. Discussion just goes mechanistically through the findings without really diving into the findings, except to state that they are supported or not by previous research. But, what are the implications? How the populations, both pets and owners will benefit? What kind of service and understanding could we bring to people with mental health issues with your findings? What about other species, would the findings hold? What about the discrepancy in the number of pets? What does it say about mental health? Generalizability to other ages? Typical populations with no mental health issues? People located elsewhere?

6. In the abstract, authors state that their sample is emerging adults with anxiety and low mood. This is a selection bias and must be addressed. In Discussion, they frame this status as a finding, i.e., these emerging adults were determined to have anxiety issues (as an after thought). Yet, methods state that they were selected because they identified as having difficulties with anxiety/depression etc. Which one is it? Also, some were unemployed. This issue should also be addressed in the discussion section.

7. I counted 12 tables, plus mediation figures. That is simply too much and is part of the problem of presenting the paper crisply and making sense of the results and discussing them. Why all mediation analyses necessary? There are many ways the info could be crisp. For example, why couldn’t each hypothesis define a very clear dependent and independent variable(s)? For example, if multiple measures are used, they could be aggregated, or an index could be constructed. One should also keep in mind that multiple tests suffer from Type I error: rejecting a true null hypothesis (false positive). The more tests you do, the more likely that you would commit this error. Moreover, some measures are likely be related causing multicollinearity. A succinct measure/index could address these problems. At the least, it would render the paper much more readable. At the least, the rationale for mediating effects should be stated.

Another approach to present the results much more crisply is to remove many of the tables and write significant statistical values in parentheses, right after the results, omitting nonsignificant ones with a statement of Ps>0.05). For ex: there was a negative correlation between X, & Y (r,.. p….one-tailed).

6. PLOS authors have the option to publish the peer review history of their article (what does this mean?). If published, this will include your full peer review and any attached files.

Reviewer #1: No

Reviewer #2: No

---

## [Author Response · Author response to Decision Letter 1]

28 Jul 2025

Response to Reviewers

Editor Comments

Comment: Please ensure that your manuscript meets PLOS ONE's style requirements, including those for file naming.

Response: We have ensured the files meet the style requirements.

Comment: We note that the grant information you provided in the ‘Funding Information’ and ‘Financial Disclosure’ sections do not match. When you resubmit, please ensure that you provide the correct grant numbers for the awards you received for your study in the ‘Funding Information’ section.

Response: These now match.

Comment: Please state what role the funders took in the study. If the funders had no role, please state: ""The funders had no role in study design, data collection and analysis, decision to publish, or preparation of the manuscript."" If this statement is not correct you must amend it as needed. Please include this amended Role of Funder statement in your cover letter; we will change the online submission form on your behalf.

Response: The funders had no role in the study, and so we have added the statement, “The funders had no role in study design, data collection and analysis, decision to publish, or preparation of the manuscript.” This has been included in the cover letter.

Comment: In the online submission form you indicate that your data is not available for proprietary reasons and have provided a contact point for accessing this data. Please note that your current contact point is a co-author on this manuscript. According to our Data Policy, the contact point must not be an author on the manuscript and must be an institutional contact, ideally not an individual. Please revise your data statement to a non-author institutional point of contact, such as a data access or ethics committee, and send this to us via return email. Please also include contact information for the third party organization, and please include the full citation of where the data can be found.

Response: The institutional contact is ethics.hiss@ed.ac.uk. The project is registered on the OSF platform (link: https://osf.io/s5ejy/) – the link has been provided within the manuscript.

Comment: Please include your full ethics statement in the ‘Methods’ section of your manuscript file. In your statement, please include the full name of the IRB or ethics committee who approved or waived your study, as well as whether or not you obtained informed written or verbal consent. If consent was waived for your study, please include this information in your statement as well.

Response: This has now been included.

Comment: Please include captions for your Supporting Information files at the end of your manuscript, and update any in-text citations to match accordingly. Please see our Supporting Information guidelines for more information: http://journals.plos.org/plosone/s/supporting-information.

Response: This has now been done.

Additional Editor Comments:

Comment: ABSTRACT 1. L-52, instead of using ‘overlooked’ in past research, I would use soft tone words.

Response: This has been changed to ‘under researched’.

Comment: INTRODUCTION 1. I understand that bring human-human relation and psychology to compare and contract the human-pet relation is important, however the introduction section is too long. I recommend to trim it down. Having said so, I think more could be introduced about dog versus cat ownership and attachment.

Response: We have now added a little more around dog vs cat ownership. We have also made the introduction more concise.

Comment: When discussing about such important subject, please provide recent data on pet ownership, cat and dog ownership stats., etc. The following two manuscripts have cited that information. https://doi.org/10.1111/1541-4337.70060
https://doi.org/10.1016/j.cvsm.2021.01.010

Response: We have now provided recent data on these statistics.

Comment: METHOD 1. L273 and L434-435-Just curious why being a UK nationality was important for this study?

Response: We were constrained by time and funding to carry out a larger cross-cultural comparison. However, by focusing on the UK, this allowed for a more in-depth and coherent analysis of human-pet relationships within a consistent cultural, legal, and social context, avoiding the complexities and variability inherent in cross-cultural comparisons. This approach also benefits from high-quality data and well-established animal welfare frameworks unique to the UK. The funders were also UK specific. We have added this line to the methods, “The focus was on the UK only to ensure a more in-depth and coherent analysis of human-pet relationships within a consistent cultural, legal, and social context, avoiding the complexities and variability inherent in cross-cultural comparisons”.

Comment: METHOD 2. How was the participants selected that have such a huge proportion of people who were having mental health difficulties? Is this a random sample or selective/targeted samples?

Response: This study specifically targeted young adults experiencing difficulties with anxiety and/or mood, as outlined in the inclusion criteria. This focus was reflected in both the recruitment materials and strategies (e.g., outreach through mental health platforms). These details are explained in the manuscript.

Comment: DISCUSSION 1. Did the study find an answer to a question on whether, the pet owners in the study owned pets to seek mental wellbeing or they realized that the pets became part of their mental health, once they started keeping pets?

Response: Given the quantitative nature of this study, it was not possible to determine whether participants acquired pets specifically for mental wellbeing purposes or whether they came to recognize the mental health benefits after pet ownership began. We have acknowledged this limitation regarding the direction of effect in the manuscript; “Furthermore, whether pets are acquired intentionally to support mental wellbeing or whether the perceived mental health benefits emerge after acquisition remains an open question. This could be more effectively explored through future qualitative research, longitudinal data collection, or prospective cohort studies.”

Reviewers' comments:

Reviewer #1

Comment: This study explored the interaction of pet attachment orientations and owner wellbeing in emerging adults owning cats or dogs, while considering the role of pet compatibility and perceived pet wellbeing and (problem) behaviour. They found an intricate interplay between the factors, with insecure pet attachment correlating differently with anxiety or depression in dog vs cat owners. Moreover, compatibility and behaviour mediated these interactions in both species-owners, while only dogs’ mental wellbeing had a mediating effect among the perceived pet wellbeing factors. The authors conclude that the dynamics of pet ownership and wellbeing are much more complicated than generally assumed, and that pet attachment might be a risk factor for wellbeing. Given that more than half of the population is estimated to own a pet and that the literature on pet ownership and wellbeing is divided, the paper tackles an important topic and provides a novel investigation by combining factors that have previously shown to influence human and animal wellbeing. Furthermore, they focus on a demographic that is particularly vulnerable to mental health challenges. The paper provides a well-structured introduction into the different factors they plan to test and the scales they used to do so. The results are likewise well structured and in combination with the figures, easy to follow. However, I point out some caveats in the interpretation of the results and the appraisal of the methods that might benefit the validity of the paper.

Response: Thank you for your time in reviewing our paper and for your useful suggestions, we believe the paper has now been improved following peer review.

Comment: In short, I see three main issues: 1) the methodology (self-reports) only allow for correlations, and while the mediation analysis has some merit, I do not feel that the conclusions and interpretations can be stated as causally as they were. For example, the results underlying the conclusion that “attachment insecurity [is] a potential risk factor for poorer owner mental health” might equally be explained by people with poorer mental health forming more insecure attachments to their pet, given that we do not know based on the questionnaire whether the animal or the mental health problems came first.

Response: We fully agree that the cross-sectional, self-report nature of the study limits our ability to make causal inferences. Our interpretation of the mediation model was intended to suggest potential pathways rather than assert causality. We have revised the manuscript to clarify that findings reflect associations rather than directional effects, and we now explicitly note that the directionality between attachment insecurity and mental health outcomes cannot be determined. We have also emphasized this limitation in the discussion and limitations sections, and have adjusted our language to avoid implying causality.

Comment: 2) the authors measure whether the participants were more or less avoidantly/anxiously attached to the animals, but there was no measure on how strong that attachment was (e.g. some score on bonding strength) or whether it was an attachment (in the deeply bonded sense of the word) at all. This might not only have differed across participants, but also between cats and dogs, given that we know dogs exhibit attachment behaviour, but cats might not (references see introduction). While this cannot be amended in hindsight, I would be grateful to hear the authors’ opinion on how this might or might not have influenced the attachment orientations, and if needed, discuss this drawback as limitation.

Response: We appreciate this important observation. It is true that the Pet Attachment Anxiety and Avoidance measure assesses attachment orientation (i.e., anxious or avoidant), rather than the strength of attachment or whether a strong attachment bond exists at all. As such, it is possible that participants with lower levels of attachment strength, or no true attachment bond, may still receive scores on this measure (noting however that it is rare in this type of research to recruit a participant without a bond to their animal), potentially affecting the interpretation of findings. Furthermore, differences in species—particularly between cats and dogs, which vary in their expression of attachment-related behaviors—could influence how attachment orientations are experienced and reported, and a limitation of this field is that most human-animal measures are biased towards human-dog relationships. While this limitation cannot be retroactively addressed, we have now added a note to the limitations section acknowledging that (1) the measure does not assess attachment strength or presence per se, and (2) species-specific behavioral differences may influence attachment dynamics and should be considered in future research.

Comment: 3) Finally, I am missing consideration for the study populations’ background factors, especially the fact that most of the people still lived at home. For one, this might suggest that many of them did not choose their pet themselves and/or have a different relationship to the pet than single/pair pet owners do (e.g. fewer caretaker responsibilities, less choice on getting a compatible animal, less involvement in training/daily interactions…). Secondly, this might mean that family environment (e.g. parenting style) might influence both the pet attachment and the owner wellbeing rather than one influencing the other directly (see details below). While they are a critical demographic, they may hence not be the most representative model to generalize the outcomes to pet ownership in general. Most points in the discussion could benefit from these considerations.

Response: We appreciate this insightful comment. We agree that the living arrangements of our study population—particularly the fact that many participants were still residing in the family home—could have influenced both the nature of their relationship with the pet and their overall wellbeing. As the reviewer notes, participants may not have chosen their pet, may have had less autonomy in caring for them, and may have engaged differently in daily interactions, all of which could shape attachment patterns. Furthermore, shared environmental factors, such as family dynamics or parenting style, could plausibly affect both pet attachment and mental health, potentially confounding observed associations. In response to this comment, we have revised the discussion to more explicitly consider these contextual factors and have acknowledged that emerging adults living at home may not represent the broader pet-owning population. We have also added this point to the limitations section to caution against overgeneralization of the findings.

Comment: Beyond that, I point out some minor issues like lacking consistency or unclear sentences below. Beside these interpretational points, the paper has tackled an important topic with sound methodology, and I see merit in its publication after the consideration of the abovementioned caveats.

Responses: Thank you for your interest in this topic and for your thoughtful responses to our findings. We have now addressed these and believe our manuscript has been improved.

Comment: General:- Please be consistent in language use, e.g. well-being and wellbeing, behaviour and behavior.

Response: We have edited the manuscript throughout to be more consistent.

Comment: Introduction - L133 more positive compared to other pets? Or more positive rather than negative? And from what I know, cats are among the most studied pets after dogs. What about bunnies, rats, birds, reptiles? If they are not of interest, I would recommend either specifying the general pet comment or adding why you focus on these two species in particular (e.g. “dogs and cats are the most frequent pets and even there, outcomes are ambiguous or lacking.”)

Response: Thank you for this comment - you’re absolutely right—our wording (“more positive compared to other pets”) may unintentionally imply a broader conclusion than our data supports. We have revised it to something more precise, “than negative outcomes.”

Regarding the focus on dogs and cats, indeed, cats and dogs receive extensive attention in the literature—but not to the exclusion of other species. For instance, after dogs and cats, rabbits are the third most common companion animals in UK households, while birds, rodents, reptiles and others follow in declining prevalence; however, wellbeing benefits of such other animals have been overlooked in research. This study therefore focused on dogs and cats because they are the most common pets owned in UK households, and feature most prominently in the empirical literature on pet attachment and human well‑being. Although smaller-scale studies exist for rabbits, birds, rodents, and reptiles, these species are far less represented in wellbeing research. Therefore, our focus on dogs and cats reflects both their prevalence and the concentration of relevant data. We have added a sentence to reflect this within the manuscript; “The focus on dogs and cats in the current study is justified both by their predominance in UK households, and by the concentration of mental‑health research on these species. Even within this domain, however, findings are far from consistent: studies report positive, null, and occasionally adverse effects on owner well‑being. This mixed evidence underscores the need for cautious interpretation of pet‑attachment benefits and supports the rationale for our focused scope.”

Comment: - L135 “Human-dog attachments are, however, bi-directional and there is reciprocity within the relationship, which is important for wellbeing [35, 36], yet less is known regarding the nature of human-cat relationships, which may be more variable, and where different attachment features may be observed [37, 38].” I have trouble following this argument. Partially, it repeats what it says above. Partially, I am confused about the “attachment features”. Do you mean it might not be bidirectional with cats? Studies diverge on w

---

## [Decision Letter · Decision Letter 1]

2 Sep 2025

PONE-D-24-52893R1
Exploring the Connection Between Pet Attachment and Owner Mental Health: The Roles of Owner-Pet Compatibility, Perceived Pet Welfare, and Behavioral Issues.
PLOS ONE

Dear Dr. Hawkins,

Thank you for submitting your manuscript to PLOS ONE. After careful consideration, we feel that it has merit but does not fully meet PLOS ONE’s publication criteria as it currently stands. Therefore, we invite you to submit a revised version of the manuscript that addresses the points raised during the review process.

**The edits and revisions have substantially improved the overall quality of the manuscript. However, both the reviewers and I feel that the Introduction and Strengths and Limitations sections remain somewhat lengthy and could be streamlined to avoid verbosity. Please address the following detailed comments provided by the two reviewers.**

We look forward to receiving your revised manuscript.

Kind regards,

Janak Dhakal

Academic Editor

PLOS ONE

**Journal Requirements:**

Reviewers' comments:

Reviewer's Responses to Questions

**Comments to the Author**

1. If the authors have adequately addressed your comments raised in a previous round of review and you feel that this manuscript is now acceptable for publication, you may indicate that here to bypass the “Comments to the Author” section, enter your conflict of interest statement in the “Confidential to Editor” section, and submit your "Accept" recommendation.

Reviewer #1: (No Response)

Reviewer #2: (No Response)

2. Is the manuscript technically sound, and do the data support the conclusions?

Reviewer #1: Yes

Reviewer #2: Yes

3. Has the statistical analysis been performed appropriately and rigorously? 

Reviewer #1: Yes

Reviewer #2: Yes

4. Have the authors made all data underlying the findings in their manuscript fully available?

Reviewer #1: Yes

Reviewer #2: Yes

5. Is the manuscript presented in an intelligible fashion and written in standard English?

Reviewer #1: Yes

Reviewer #2: Yes

6. Review Comments to the Author

**Reviewer #1:** The authors have done a remarkable job in improving the manuscript. The introduction is still on the long side, but understandable. The discussion is a lot clearer and more nuanced, though some of the paragraphs would benefit from more conciseness and alternative explanations. I note a few minor points below and would be happy to see this important work published once these are addressed.

Abstract

L48 delete comma

L53 “Attachment notably influenced mental health” I find this phrasing still too causal given the correlative results. It has been amended well in the rest of the sentence, but this part could use a more nuanced version as well to not misguide the reader.

L61 “Owner-dog compatibility, particularly in the affection domain, influenced owner anxiety” same as above.

Introduction

L141 putting “negative outcomes” in brackets rather than integrated it in a sentence makes it a bit awkward to read. If possible, I would suggest integrating it as a normal part of the sentence.

L154 ff this paragraph still feels rather repetitive and could be a lot more concise.

Methods

L264 something is off in this sentence (Prolific appears twice)

L329 and L341: delete “is”

Results

L430 f It is unclear what these outcomes compare to. More likely than dog owners, or more likely to display insecure than secure attachment?

L481 delete the comma before subscales?

L480ff these outcomes are surprising. A negative relationship between insecure attachment and cat healthy behaviours but also a negative relationship between insecure attachment and cat clinical signals? How could this be explained?

L543 closing bracket is missing

L605 two periods

Discussion

L645 you write under researched without a hyphen in the abstract

L645 “are believed” by whom? Did you hypothesise that or can you accredit this assumption to someone?

L761 I am unsure what the “or as not” is meant to express. Perhaps delete the comma afterwards?

L769 ff higher total quality of life and physical functioning relate to the pet, correct? I would suggest adding “dogs’ ” in front of the terms to help the reader stay on track with all the different variables and associations. The same goes for the rest of the paragraph. It is at times difficult to follow when a term relates to the dog or the owner without having to go back to the exact analysis.

L831 “These findings indicate that insecurity within human-pet attachments could influence feelings of owner compatibility and thereby perceptions of the human-pet relationship”. Or the other way around (which I would personally find a more straight forward explanation)? Feeling incompatible may lead to insecure attachment – again considering what kind of demographic (adolescents at home that might not have chosen the pet/ have a thorough attachment)

L856 the strength and limitation section is rather long. I believe parts of that could be significantly more succinct. For example, L856 – 871: the strength could be summarized in one sentence, since this has already been discussed in the intro. Sentence 868 is not needed here, it fits better in the conclusion.

L919 in the table you did not at the Q to LGBTQI+

Graphs: The graphs should be higher quality, they are hard to read.

**Reviewer #2:** I acknowledged that the manuscript has significantly improved, however, it still struggles to present the research questions with clarity and precision.

Specifically, the introduction is still too long, the research questions are repetitive, and predictions are missing. It appears the authors inadvertently introduced this issue in an effort to address my comments regarding one-to-one correspondence between research hypotheses and results. Perhaps, I can make some suggestions- this issue alone does not preclude publication.

The authors are asking

1) Are there differences between dog and cat owners on measures of pet attachment and mental health?, 2) Does insecure pet attachment relate to owner mental health symptom severity?, 3) Does perceived pet welfare explain the relationship between insecure pet attachment and owner mental health symptom severity?, 4) Do perceived pet behavioral problems explain the relationship between insecure attachment and owner mental health symptom severity?, and 5) Does owner-pet compatibility explain the relationship between insecure attachment and owner mental health symptom severity?

How about presenting it in the following manner:

1) Are there differences between dog and cat owners on measures of pet attachment and mental health?

2) Do insecure pet attachment, perceived pet welfare and behavioral problems, and owner-pet compatibility explain the relationship between insecure attachment and owner mental health symptom severity?

This will also leave the breakdown in the results section perfectly fine.

The authors should then introduce these questions around line 150, right after …present research… The rest would constitute the rationale for predictions, which are missing by the way. Where are the predictions following each research question?

In sum, introduce the significance of the study, articulate the gap in our current knowledge, present the specific research questions, and follow each with predictions grounded in a clear rationale supported by prior research.

A minor issue: In discussion, this sentence: “These aspects, which are under-researched, are believed to play crucial roles in shaping owner-pet relationships and consequently influence owner mental well-being” (Line 645-47) requires references. Additionally, consider revising it to use active voice rather than passive.

7. PLOS authors have the option to publish the peer review history of their article (what does this mean?). If published, this will include your full peer review and any attached files.

Reviewer #1: No

Reviewer #2: No

---

## [Author Response · Author response to Decision Letter 2]

15 Sep 2025

Reviewer #1

Comment: The authors have done a remarkable job in improving the manuscript. The introduction is still on the long side, but understandable. The discussion is a lot clearer and more nuanced, though some of the paragraphs would benefit from more conciseness and alternative explanations. I note a few minor points below and would be happy to see this important work published once these are addressed.

Response: Thank you for your positive feedback. In line with both reviewer’s and editors’ feedback, we have further trimmed down the Introduction to make this more succinct, as well as the Strengths and Limitations sections to be more concise, and have addressed your final feedback comments below.

Comments:

Abstract

L48 delete comma

L53 „Attachment notably influenced mental health” I find this phrasing still too causal given the correlative results. It has been amended well in the rest of the sentence, but this part could use a more nuanced version as well to not misguide the reader.

L61 “Owner-dog compatibility, particularly in the affection domain, influenced owner anxiety” same as above.

Response: These edits have now been addressed.

Introduction

Comment: L141 putting “negative outcomes” in brackets rather than integrated it in a sentence makes it a bit awkward to read. If possible, I would suggest integrating it as a normal part of the sentence.

Response: This has been re-worded.

Comment: L154 ff this paragraph still feels rather repetitive and could be a lot more concise.

Response: This paragraph has been edited to be more concise.

Comments:

Methods

L264 something is off in this sentence (Prolific appears twice)

L329 and L341: delete “is”

Response: These edits have now been addressed.

Results

Comment: L430 f It is unclear what these outcomes compare to. More likely than dog owners, or more likely to display insecure than secure attachment?

Response: More likely than dog owners - this has been re-worded for clarity.

Comment: L481 delete the comma before subscales?

Response: Have re-worded this sentence as we meant subscales of the FHQLS, and not the direct quality of life scale.

Comment: L480ff these outcomes are surprising. A negative relationship between insecure attachment and cat healthy behaviours but also a negative relationship between insecure attachment and cat clinical signals? How could this be explained?

Response: Yes, you are right that insecure attachment led to perceptions of poorer cat welfare, yet fewer clinical signs of health issues. This may highlight the difference between subjective welfare/quality of life reports and owner-reported clinical signs. Perceptions of welfare are inherently subjective and require interpretation of the cat’s emotional state, satisfaction, or “happiness.” Owners with insecure attachments may have a negative cognitive bias that influences these judgments (explained further in the discussion section). Clinical signs, in contrast, are more objective (e.g., vomiting, limping, coughing) and less open to interpretive bias. Even if insecurely attached owners evaluate welfare more negatively, they may not report increased clinical signs because those are observable, concrete events rather than perceptions. We have now added a sentence to this effect in the discussion.

Comments:

L543 closing bracket is missing

L605 two periods

Response: Edited.

Discussion

Comment: L645 you write under researched without a hyphen in the abstract

Response: Edited.

Comment: L645 “are believed” by whom? Did you hypothesise that or can you accredit this assumption to someone?

Response: Edited.

Comment: L761 I am unsure what the “or as not” is meant to express. Perhaps delete the comma afterwards?

Response: Edited.

Comment: L769 ff higher total quality of life and physical functioning relate to the pet, correct? I would suggest adding “dogs’ ” in front of the terms to help the reader stay on track with all the different variables and associations. The same goes for the rest of the paragraph. It is at times difficult to follow when a term relates to the dog or the owner without having to go back to the exact analysis.

Response: Thank you for this suggestion – this has now been edited for clarity throughout.

Comment: L831 “These findings indicate that insecurity within human-pet attachments could influence feelings of owner compatibility and thereby perceptions of the human-pet relationship”. Or the other way around (which I would personally find a more straight forward explanation)? Feeling incompatible may lead to insecure attachment – again considering what kind of demographic (adolescents at home that might not have chosen the pet/ have a thorough attachment)

Response: Thank you for this insightful comment; we agree that the direction could plausibly work both ways and we have clarified in the discussion that given the cross-sectional mediation design, causality cannot be determined and the relationship between insecure attachment and perceived compatibility is likely bidirectional.

Comment: L856 the strength and limitation section is rather long. I believe parts of that could be significantly more succinct. For example, L856 – 871: the strength could be summarized in one sentence, since this has already been discussed in the intro. Sentence 868 is not needed here, it fits better in the conclusion.

Response: We have now edited this section to be more concise.

Comment: L919 in the table you did not at the Q to LGBTQI+

Response: I could see the Q in the table (demographics); however I did notice we missed the Q on a separate mention so this has been edited.

Comment: Graphs: The graphs should be higher quality; they are hard to read.

Response: Thank you – we have tried to improve the quality of these, increasing the KBs of each.

Reviewer #2:

Comments:

I acknowledged that the manuscript has significantly improved, however, it still struggles to present the research questions with clarity and precision. Specifically, the introduction is still too long, the research questions are repetitive, and predictions are missing. It appears the authors inadvertently introduced this issue in an effort to address my comments regarding one-to-one correspondence between research hypotheses and results. Perhaps, I can make some suggestions- this issue alone does not preclude publication. The authors are asking 1) Are there differences between dog and cat owners on measures of pet attachment and mental health?, 2) Does insecure pet attachment relate to owner mental health symptom severity?, 3) Does perceived pet welfare explain the relationship between insecure pet attachment and owner mental health symptom severity?, 4) Do perceived pet behavioral problems explain the relationship between insecure attachment and owner mental health symptom severity?, and 5) Does owner-pet compatibility explain the relationship between insecure attachment and owner mental health symptom severity? How about presenting it in the following manner: 1) Are there differences between dog and cat owners on measures of pet attachment and mental health?, 2) Do insecure pet attachment, perceived pet welfare and behavioral problems, and owner-pet compatibility explain the relationship between insecure attachment and owner mental health symptom severity?

This will also leave the breakdown in the results section perfectly fine. The authors should then introduce these questions around line 150, right after …present research… The rest would constitute the rationale for predictions, which are missing by the way. Where are the predictions following each research question? In sum, introduce the significance of the study, articulate the gap in our current knowledge, present the specific research questions, and follow each with predictions grounded in a clear rationale supported by prior research.

Response: We really appreciate the time and effort you have put in to help us improve the manuscript – we have taken on your suggestions and have re-worded/ordered the research questions, and included the predictions for each. We have cited theory and research to support each prediction, but did not want to repeat the rationales provided throughout the introduction section. We have kept the order and flow of the results the same but have removed the research question numbering. We hope this has made the flow clearer and avoids repetition. We have also now made the introduction, and some sections of the discussions more succinct in line with the other reviews.

Comment: A minor issue: In discussion, this sentence: “These aspects, which are under-researched, are believed to play crucial roles in shaping owner-pet relationships and consequently influence owner mental well-being” (Line 645-47) requires references. Additionally, consider revising it to use active voice rather than passive.

Response: This has been edited, and relevant citations have been included to support this statement.

---

## [Editor Report · Decision Letter 2]

18 Sep 2025

Exploring the Connection Between Pet Attachment and Owner Mental Health: The Roles of Owner-Pet Compatibility, Perceived Pet Welfare, and Behavioral Issues.

PONE-D-24-52893R2

Dear Dr. Hawkins, 

We’re pleased to inform you that your manuscript has been judged scientifically suitable for publication and will be formally accepted for publication once it meets all outstanding technical requirements.

Kind regards,

Janak Dhakal

Academic Editor

PLOS ONE
---

## [Editor Report · Acceptance letter]

PONE-D-24-52893R2

PLOS ONE

Dear Dr. Hawkins,

I'm pleased to inform you that your manuscript has been deemed suitable for publication in PLOS ONE. Congratulations! Your manuscript is now being handed over to our production team.

Kind regards,

on behalf of

Dr. Janak Dhakal

Academic Editor

PLOS ONE